# ONLY BRAINS ALIGN WITH BRAINS: CROSS-REGION ALIGNMENT PATTERNS EXPOSE LIMITS OF NORMATIVE MODELS

**Larissa Höfling**[1,*]     **Matthias Tangemann**[2,*]     **Lotta Piefke**[2]     **Susanne Keller**[2]
**Katrin Franke**[1,3,†]     **Matthias Bethge**[2,†]
[1]**University Clinics Tübingen**     [2]**University of Tübingen, Tübingen AI Center**
[3]**Byers Eye Institute, Stanford Medicine**
`{larissa.hoefling, katrin.franke}@uni-tuebingen.de`

## ABSTRACT

Neuroscientists and computer vision researchers use model–brain alignment benchmarks to compare artificial and biological vision systems. These benchmarks rank models according to alignment measures such as the similarity of representational geometry or the predictability of neural responses from model activations. However, recent works have identified a number of problems with these rankings, among them their lack of discriminative power and robustness, raising the conceptual question of what it means for a model to be 'brain-aligned'. Here we introduce *alignment patterns* - characteristic functional relationship profiles of each brain region to all others - and propose that models should reproduce these patterns to qualify as brain-aligned. First, we apply a standard benchmarking pipeline to a broad spectrum of vision models on the BOLD Moments video fMRI dataset across visual regions of interest (ROIs). We find diverse models appear *equivalent* in their brain alignment, reflecting the lack of discriminative power of conventional alignment benchmarking pipelines based on pointwise, i.e. ROI-layer, comparisons. In contrast, *alignment pattern analysis (APA)* is a second-order structural consistency test: a model aligned to a given ROI should reproduce that ROI's characteristic cross-region alignment profile. Applying APA, we find that, while these patterns are highly stable across brains of different subjects, even top-ranked models often fail to capture them. Notably, models that appear effectively equivalent in their pointwise alignment to an ROI diverge sharply under the relational criterion, demonstrating the added discriminative value of APA. Finally, we argue for a clearer distinction between the criteria a model must meet to serve as a tool versus as a computational model for human visual cortex. Conventional alignment measures may be sufficient for identifying neurally predictive models, but claims about computational or algorithmic similarity may require a stronger basis of evidence, including the reproducibility of relational alignment patterns.

Code is available at https://github.com/bethgelab/alignment-pattern-analysis.

## 1 INTRODUCTION

Vision science aims at understanding the computational principles that support robust visual behaviors in primates and other animals. During the last decade, DNNs have become an increasingly important tool towards this goal since they approach visual capabilities of humans and are highly predictive of neural activity (Yamins et al., 2014; Yamins & DiCarlo, 2016). This has led to a new research paradigm, where diverse collections of computer vision models are evaluated in terms of *alignment* with functional recordings from visual brain areas using a variety of measures (Klabunde et al., 2025; Barbosa et al., 2025). Correlations of model-brain alignment with key model parameters, such as the training dataset, objective and architecture, then hint at biologically relevant constraints (Tang et al., 2025; Conwell et al., 2024; Sartzetaki et al., 2024). Recent progress in computer vision, including performant self-supervised models (e.g. Bear et al., 2023; Assran et al.,

---

[*,†] Equal contribution

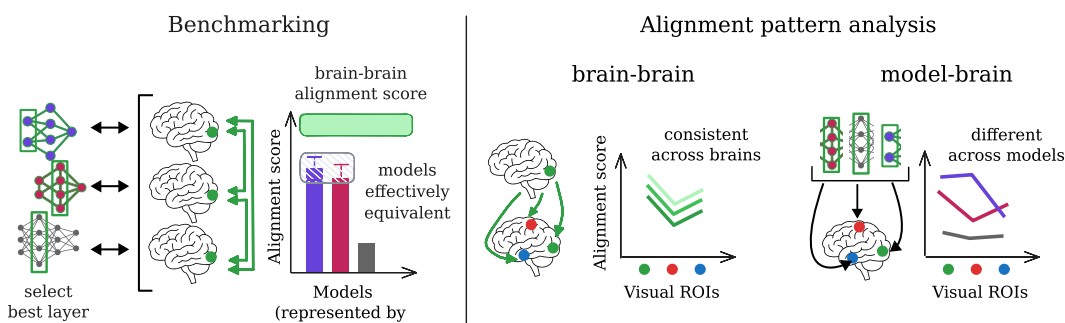

Figure 1: **Alignment pattern analysis to distinguish between equivalently aligned models. Left:** Standard brain-alignment benchmarks rank models according to their alignment to a brain region under some similarity transform. Comparing models' alignment scores to a reference derived from brain-brain alignment scores aids interpreting model scores (e.g. the NeuroAI Turing Test Feather et al., 2025), but leaves open the problem of a lack in discriminative power to distinguish between equivalently aligned models. **Right:** We introduce alignment pattern analysis (APA) as a relational extension to the NeuroAI Turing Test that distinguishes models based on whether they reproduce the characteristic cross-region alignment patterns of brains.

2025), along with the systematic collection of large-scale fMRI datasets (Lahner et al., 2024; Allen et al., 2022), now make it possible to scale these efforts towards comprehensive brain-alignment benchmarks of vision models (Schrimpf et al., 2018; 2020; Conwell et al., 2024; Sartzetaki et al., 2024; Tang et al., 2025).

The conclusions drawn from such benchmarks, however, rest on the assumption that differences in alignment scores reflect differences in brain–likeness in some meaningful way. Recent work has challenged the validity of this inference step on the basis of several arguments and empirical findings. Different alignment measures yield inconsistent model rankings (Soni et al., 2024; Ahlert et al., 2024), evidently measuring different aspects of alignment, but the choice of alignment measure is often not explicitly motivated. Models varying in many aspects were found to achieve similar brain-alignment scores, raising the question of whether currently used alignment measures have sufficient discriminative power to tell apart more from less brain-like models (Conwell et al., 2024; Wu et al., 2025). Further, inferences about brain-likeness from brain-alignment are usually based on correlational evidence and are rarely subjected to rigorous causal testing (Bowers et al., 2022; Schaeffer et al., 2024; Bowers et al., 2023). This raises a deeper conceptual question: What does it actually mean for a model to be "brain-aligned", and how do we measure it? Previous works (Feather et al., 2025) have posited the following requirement for a model to be considered brain-like: its internal representations should be indistinguishable - under a similarity transform - from those of other brains up to individual variability. In other words, if a model resembles a brain as much as one brain resembles another, it passes the *NeuroAI Turing Test* (Fig. 1, left).

In our work, we adopt the viewpoint of the NeuroAI Turing Test and make three major contributions to advance the benchmarking of model-brain alignment:

1. **Benchmarking vision models on a video fMRI dataset**
   We evaluate a broad family of state-of-the-art vision models with respect to their alignment to brain activations across human visual cortex to naturalistic videos and find that diverse models achieve comparable alignment scores (Fig. 2). This finding suggests a lack of discriminative power of conventional alignment benchmarks.

2. **Operationalising equivalence in brain-alignment**
   We quantify this by devising a heuristic based on alignment score distributions across subjects to define *effective equivalence* (Fig. 1, left). Using this heuristic, we find that many distinct models are equivalent in their brain alignment, confirming the need for more discriminative practices in alignment benchmarks.

3. **Alignment pattern analysis to distinguish among equivalently aligned models**
   Third, we introduce a *relational* criterion to distinguish among equivalently aligned mod-

els: We posit that a model should only be considered aligned to a brain region if it reproduces the *cross-region alignment pattern* of this brain region (Fig. 1, right).

## 2 RELATED WORK

**Alignment benchmarks.** Pioneered by work on visual object recognition (Yamins et al., 2014), neural network models have been compared to the brain on large-scale benchmarks. Brain-Score (Schrimpf et al., 2018) originally focused on static image processing along the ventral stream, later adding language regions (Schrimpf et al., 2021). It provided a first large-scale platform for model evaluation. The Algonauts challenge extended this approach to whole-brain responses to naturalistic stimuli, first for static images and later extending to dynamic movie-based paradigms (Gifford et al., 2024; 2023; Cichy et al., 2021; 2019). Most recently, the challenge has emphasized multimodal video inputs, pushing alignment analyses into richer and more ecologically valid contexts (see e.g. d'Ascoli et al., 2025).

Beyond these community benchmarks, several studies have systematically examined factors shaping model–brain alignment. Conwell et al. (2024) showed that vastly different architectures can achieve similar alignment, with variety of "visual diet" emerging as the most consistent determinant. Tang et al. (2025) found that a single predictive objective - understanding world dynamics - leads to high linear predictivity (LP) scores across cortical areas, suggesting shared computational principles across the hierarchy. Using representational similarity analysis (RSA), Sartzetaki et al. (2024) found that video models achieve the highest alignment in early visual regions, whereas semantic training objectives are key for both ventral and dorsal regions.

**Frameworks for tightening definitions of brain-alignment.** Standard alignment measures such as LP (Yamins et al., 2014) and RSA (Kriegeskorte et al., 2008)) have been criticized as being too flexible and hence lacking in discriminative power (Conwell et al., 2024). To address these shortcomings, stricter alignment measures (reviewed in Klabunde et al., 2025) and aggregation of several measures (Wu et al., 2025) have been proposed. Beyond this, several frameworks have been developed that aim to conceptually tighten the definition of brain-alignment. Nonaka et al. (2021) propose the brain hierarchy score to quantify hierarchical correspondence between deep neural networks and brains. Thobani et al. (2025) suggest inter-animal transforms (IACT) as a way for identifying alignment measures that exhibit a sufficient degree of specificity while not being prohibitively strict. Here, we suggest alignment pattern analysis (APA), which is complementary to these previous works. Nonaka et al. (2021) evaluate hierarchical correspondence between entire models and the entire visual stream, whereas we focus on comparing patterns of alignment of individual model layers to those of individual ROIs (Fig. 1). The IACT serves to evaluate and identify appropriate alignment measures, whereas APA can be performed with any alignment measure. Rather, APA can be seen as a relational extension to the NeuroAI Turing Test Feather et al. (2025), which suggests to compare model-brain alignment scores with brain-brain alignment scores; APA compares model-brain with brain-brain alignment *patterns*.

## 3 METHODS

### 3.1 DATASET

We base our analyses on the BOLD Moments Dataset (Lahner et al., 2024), a 3T fMRI dataset recorded from 10 subjects watching over 1000 different 3-second video clips. Each of the 1000 stimuli in the train split was shown three times to each subject, each of the 102 stimuli in the test split was shown ten times. Stimulus repetitions were presented in random order across 4 sessions. We use beta values (GLMSingle regression coefficients of each voxel and video shown), projected to fslr32k surface space, as they are output from the preprocessing pipeline (specifically version B, see Lahner et al., 2024, for details). We use the original train-test split and average the fMRI activity over repetitions, leading to a higher signal-to-noise ratio on the test split, compared to the train split. While the voxel-wise beta values provided by Lahner et al. (2024) are already centered and normalized across individual sessions, we normalize and center them once more across the train set, and use the same standard deviation and mean per feature to approximately center and normalize the test split.

We analyze ROIs from the Glasser HCP-MMP atlas (Glasser et al., 2016): early visual areas (V1,V2,V3), dorsal stream and MT+ complex (V3A, V3B, V6, V6A, V7, IPS1, MST, MT, FST, LO1–LO3), and ventral stream (V4, V8, PIT, FFC; Fig. S3.1).

### 3.1.1 REFERENCE ALIGNMENT SCORES FROM INTERSUBJECT CONSISTENCY

We compute two estimates of intersubject consistency for each ROI in the following ways: **Upper estimate of intersubject consistency.** We average the fMRI data of N-1 subjects for the given ROI. Then we use this average as predictor feature space and compute RSA/LP score between the average and the remaining subject's ROI data. **Lower estimate of intersubject consistency.** As suggested in the Neuro-AI Turing test Feather et al. (2025), we compute an estimate of intersubject consistency based on *pairwise* alignment scores between subjects. For a given ROI, we sample five other subjects per target subject, excluding previously sampled pairs, for a total of 50 subject pairs. For each pair we compute the RSA/LP score between the regions' fMRI features of the two subjects. For an estimate of the variability of intersubject consistency across subjects, we do not divide by the number of samples (50), as they are not all independent. Instead, we divide by the number of independent predictors/prediction targets, i.e. subjects (10).

### 3.2 MODELS

We evaluate 47 state-of-the-art pretrained image and video deep learning models that cover a broad range of architectures, objectives and datatsets:

**Taskonomy model bank.** A collection of 26 models based on ResNet-50 and trained on the same dataset of 4 million indoor scenes, but for different tasks (Sax et al., 2018; Zamir et al., 2018). **Supervised image models.** We include ResNet (He et al., 2016) and ConvNext (Liu et al., 2022b) models from the timm library (Wightman, 2019), all trained for object recognition on ImageNet-1K. **Self-supervised image models.** As counterpart to the supervised image models, we include ResNets trained on ImageNet-1K with the self-supervised SimCLR objective (Chen et al., 2020), as provided by VISSL (Goyal et al., 2021). **CLIP.** We consider the ResNet-50 and Vision Transformer based CLIP models from the original codebase, all trained to align image and text representation on a large dataset of 400M image-text pairs (Radford et al., 2021). **Supervised video models.** We use three video transformers from the mmaction2 toolkit (Contributors, 2020): MViT (Li et al., 2022), Video Swin Transformer (Liu et al., 2022a), TimeSformer (Bertasius et al., 2021). All models were trained for action recognition on the Kinetics-400 dataset. **Unsupervised video models.** We include the ViT-based counterfactual world model (CWM) (Stojanov et al., 2025) which was trained on Kinetics-400 using an adapted MAE objective (He et al., 2022). Further, we consider the V-JEPA 2 model (Assran et al., 2025) that was trained on a large-scale video dataset using a variant of the MAE objective in feature space. **VGG Transformer (VGGT).** We include the 3D foundation model VGGT (Wang et al., 2025) as comparison to the dominantly semantic models described above. This ViT-based model was trained to simultaneously predict multiple key 3D attributes from a variable number of views of a scene.

For all models, we extract representations for the last layer of up to 15 blocks (e.g., a residual block in a ResNet). For models with more blocks, we use 15 equally spaced blocks. We apply image models to each frame individually, and video models and VGG-T to the entire video clip (3s), and average representations over time. The resulting feature vectors are reduced using sparse random projection (Achlioptas, 2003) to 5919 dimensions, following the Johnson-Lindenstrauss Lemma with an epsilon of 0.1 (Achlioptas, 2001).

### 3.3 MEASURING MODEL-BRAIN ALIGNMENT

For every combination of model and ROI, we select the best layer on the training set by averaging the alignment scores over subjects. Using the selected layer for all subjects, we then report alignment scores on the test set. We consider the following two alignment metrics:

**Representational Similarity Analysis (RSA)** compares representations based on representational dissimilarity matrices (RDMs), which are sufficient statistics for the representational geometry of a system (Kriegeskorte & Wei, 2021; Kriegeskorte et al., 2008). RDMs are constructed for the model and brain representation by computing the pairwise correlation distances of the representation

($1 -$ Pearson correlation) for all samples. The overall RSA alignment score is then the Pearson correlation between brain and model RDMs.

**Linear predictivity (LP)** measures alignment by fitting a linear model that predicts brain activity from model features (e.g., Yamins et al. (2014)). We fit ridge regression models predicting the preprocessed fMRI signals on the training set using 5-fold cross-validation. We use the RidgeCV implementation from the scikit-learn package (Pedregosa et al., 2011), which selects the optimal alpha value using leave-one-out cross-validation from 19 candidate values on a logarithmic scale spanning $10^{-9}$ to $10^9$. Given the respective linear models fitted on the training set, we report the coefficient of determination ($R^2$) on the test set.

## 3.4 Determining effective equivalence between models

To determine when models are effectively equivalent in terms of brain alignment, we devise a heuristic based on bootstrap estimates of variability in model-brain alignment scores. For a given model $m$ with feature space $X_m$, we generate a bootstrap distribution of mean brain-alignment scores under a measure $\mathcal{M}$ by resampling subject indices with replacement. Specifically, we define a bootstrap index vector

$$I^* = (i_1^*, \ldots, i_{10}^*), \qquad i_k^* \sim \text{Unif}(I) \text{ with replacement}, \quad I = \{1, \ldots, 10\},$$

and compute the corresponding bootstrap estimate of the mean alignment score as

$$\frac{1}{10} \sum_{k=1}^{10} \mathcal{M}\big(X_m, Y_{i_k^*}\big).$$

We then derive 95% confidence intervals for the model's mean brain-alignment score from the resulting distribution. A model $m$ is deemed effectively equivalent to the top-ranking model $t$ if its empirical mean alignment score $\langle \mathcal{M}(X_m, Y_i) \rangle_i$ falls within the 95% confidence interval of the top ranking model.

## 3.5 Alignment Pattern Analysis

We define an alignment pattern $\alpha$ under a similarity transform $\mathcal{M}$ between a predictor feature space $\phi_p$ and $N$ target feature spaces $\Psi_t = [\psi_t^1, ..., \psi_t^N]$ as

$$\alpha(\phi_p, \Psi_t) = [\mathcal{M}(\phi_p, \psi_t^1), \mathcal{M}(\phi_p, \psi_t^2), ..., \mathcal{M}(\phi_p, \psi_t^N)] \tag{1}$$

### 3.5.1 fMRI-derived alignment patterns

For fMRI-derived alignment patterns, both the predictor and the target feature spaces are sourced from brain activity from the BOLD Moments Dataset. fMRI-derived alignment patterns are defined between pairs of subjects $p, t$, where the brain activity of subject $p$ functions as the predictor feature space $\phi_p$ and the brain activity of subject $t$ functions as the target feature space $\Psi_t$. The alignment pattern for a given ROI $r \in \{1, 2, ..., N\}$ and a pair of subjects $p, t$ is then defined as

$$\alpha_r(\phi_p, \Psi_t) = [\mathcal{M}(\phi_p^r, \psi_t^1), \mathcal{M}(\phi_p^r, \psi_t^2), ..., \mathcal{M}(\phi_p^r, \psi_t^r), ..., \mathcal{M}(\phi_p^r, \psi_t^N)] \tag{2}$$

We detail in the Appendix Section S2.1 how the variance of fMRI-derived alignment patterns is estimated.

### 3.5.2 Model-derived alignment patterns

For model-derived alignment patterns, the predictor feature space is defined as the activations in one layer $l$ of the model, $\phi_m^l$, and the target feature spaces are analogous to the case of fMRI-derived alignment patterns. A model-derived alignment pattern between model $m$ and subject $t$ for ROI $r$ is then

$$\alpha_l(\phi_m, \Psi_t) = [\mathcal{M}(\phi_m^l, \psi_t^1), \mathcal{M}(\phi_m^l, \psi_t^2), ..., \mathcal{M}(\phi_m^l, \psi_t^N)] \tag{3}$$

### 3.5.3 STRUCTURAL CONNECTIVITY-DERIVED ALIGNMENT PATTERNS

For comparing alignment patterns to structural connectivity patterns, we use a connectivity matrix based on diffusion-weighted tensor imaging (DTI) Pierpaoli et al. (1996) streamline-density from the Human Connectome Young Adult full dataset (Caron & Pestilli, 2023) as provided through brainlife Hayashi et al. (2024) (provided as 'conmat' datatype). The procedure fits streamlines - white-matter trajectory candidates Smith et al. (2012) - to diffusion MRI data. The number of streamlines intersecting both ROIs of a pair of regions is divided by the volume of both regions to obtain the 'density'-based connectivity matrix we use. For more information, see Hayashi et al. (2024), section *dMRI processing*. We average the connectivity matrices of 1065 subjects to obtain a single connectivity matrix, $C = (c_{r,t})_{r,t=1\cdots N}$ where $c_{r,t}$ is the streamline density between regions $r$ and $t$. The structural connectivity-derived alignment pattern for a given ROI $r$ is then

$$\alpha_{struct}(r) = [c_{r,1} , \ldots , c_{r,r-1} , c_{r,r+1} , \ldots , c_{r,N}] \tag{4}$$

where we exclude the ROI $r$ since self-similarity is not defined for streamline-density as alignment measure.

### 3.5.4 ALIGNMENT PATTERN SIMILARITY

Alignment pattern similarity between two alignment patterns, e.g. an fMRI-derived alignment pattern $\alpha_r(\phi_p, \Psi_t)$ and a model-derived alignment pattern $\alpha_l(\phi_m, \Psi_t)$ is calculated as

$$\rho(\alpha_r(\phi_p, \Psi_t), \alpha_l(\phi_m, \Psi_t)) \tag{5}$$

where $\rho$ is Pearson's correlation coefficient.

## 4 RESULTS

### 4.1 BENCHMARKING ALIGNMENT OF VISION MODELS TO THE VISUAL CORTEX

We evaluated a broad range of vision models with respect to their alignment to human visual cortex (including early, ventral, and dorsal regions) using two complementary alignment measures: RSA and LP. The models varied in architecture (CNNs and Transformers), training objective (various supervised and self-supervised objectives), modality (image and video), as well as model size and training dataset (Methods 3.2).

Consistent with previous work (Tang et al., 2025), we found that the self-supervised V-JEPA 2 model family (Assran et al., 2025) achieved the highest overall alignment scores across visual cortex, according to both RSA and LP (Fig. 2a, Tables S3.1, S3.3[1]). Notably, and consistent with another body of previous work (Conwell et al., 2024), the best aligned models included CLIP with a ResNet-50 backbone and a ViT backbone, and the VGG-Transformer - which differ in several important aspects from V-JEPA 2 and each other, such as the training data, training objective and overall architecture. The analysis of model performances at the level of individual ROIs agreed with this finding (Fig. 2b).

To quantify the observation that different models do similarly well, we defined a heuristic for *effective equivalence to the best-aligned model* (see Methods Section 3.4), and found several effectively equivalent models for most ROIs (Fig. 2b, bold bars). These findings demonstrated the lack of discriminative power of alignment-measure based model rankings, motivating the need for additional criteria to distinguish between models with effectively equivalent brain-alignment.

To contextualize absolute alignment scores, we further compared model–brain alignment scores to the distribution of intersubject (i.e., brain–brain) alignment scores, as recently proposed under the NeuroAI Turing Test framework (Feather et al., 2025, Fig. 2b, dash-dotted lines; Methods Section 3.1.1). We found the best-ranking models to reach or even surpass brain-brain alignment when evaluated with LP, which indicates that the benchmark is saturated for LP on almost all ROIs (Fig. 2b and Tables S3.4, S3.2). We did not observe similar signs of saturation for RSA, so that we focus on this alignment metric for the remainder of our analyses but report results for LP in the Supplementary Material.

---

[1]Models from the Taskonomy family consistently underperformed all other models in terms of brain alignment. We report scores for these models in the Tables in the Supplementary Material.

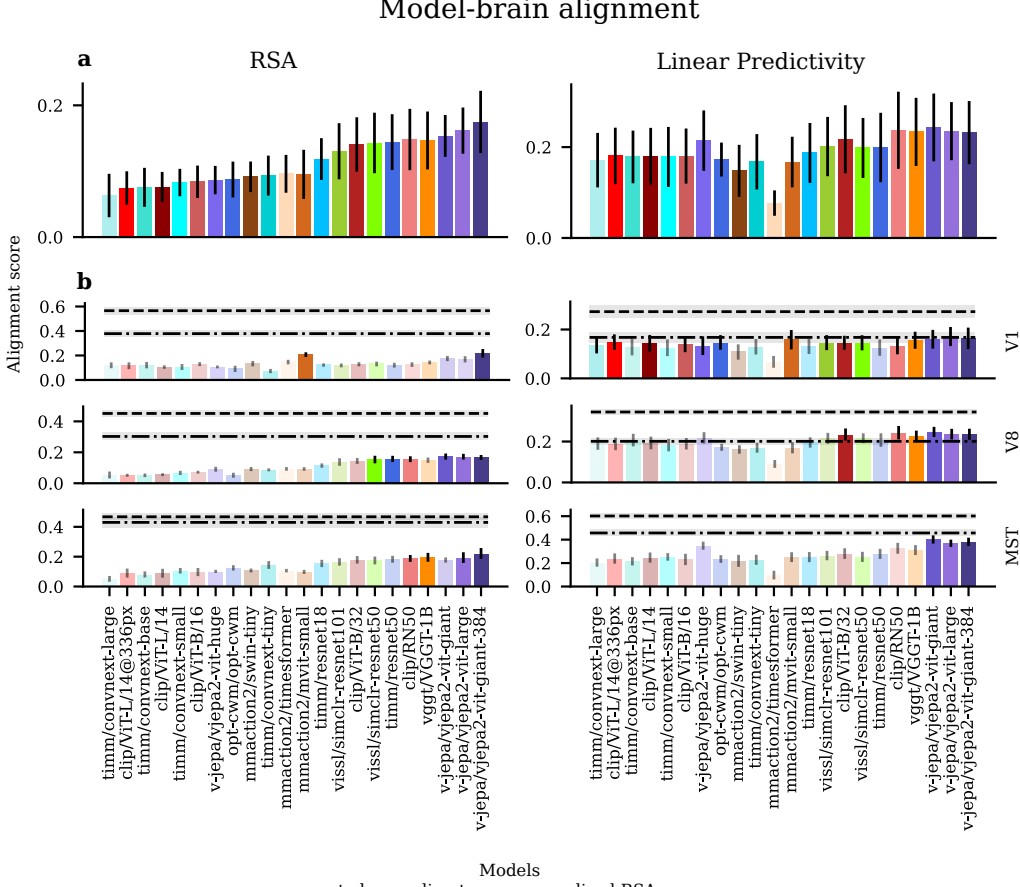

Figure 2: **Diverse models achieve comparable alignment scores on the BOLD Moments Dataset**. **(a)** Subject-averaged alignment scores (RSA/LP) across ROIs; errorbars are standard deviation across ROIs. **(b)** Subject-averaged alignment scores for individual ROIs (V1, V8, MST); errorbars indicate bootstrapped 95% confidence-intervals around the mean. Effectively equivalent models (see Methods Sec. 3.4) are highlighted in bold. Dash-dotted line indicates NeuroAI Turing Test alignment reference, i.e. pairwise alignment between individual brains. Dashed line indicates an alternative alignment reference based on comparing an averaged reference brain to individual brains (see Methods Sec. 3.1.1).

## 4.2 Alignment patterns are a reproducible fingerprint of each brain region

Visual brain areas, as functionally defined entities, have characteristic functional relationships to all other regions. We hypothesized that these relationships are reflected in stable, ROI-specific patterns of inter-region alignment - *fMRI-derived alignment patterns* (AP)[2] - that are consistent across individuals and characteristic for each ROI, and that these patterns could serve as a reference against which to evaluate models.

We estimated fMRI-derived AP by using one subject's brain activity in a given ROI as a predictor of another subject's brain activity across all ROIs (Methods Sec. 3.5); both fMRI-derived and model-derived AP use scores normalized to the lower estimate of intersubject consistency. We quantified the consistency of these patterns across the population using *alignment pattern similarity (APS)*: for each subject, we computed a reference AP from all pairwise combinations of the remaining subjects, and measured how well individual APs correlated with this reference (Suppl. Methods Sec.S2.1).

---

[2]We use AP as abbreviation for both the singular *alignment pattern* and the plural *alignment patterns* to avoid confusion with APS (alignment pattern similarity)

Brain-brain alignment pattern analysis

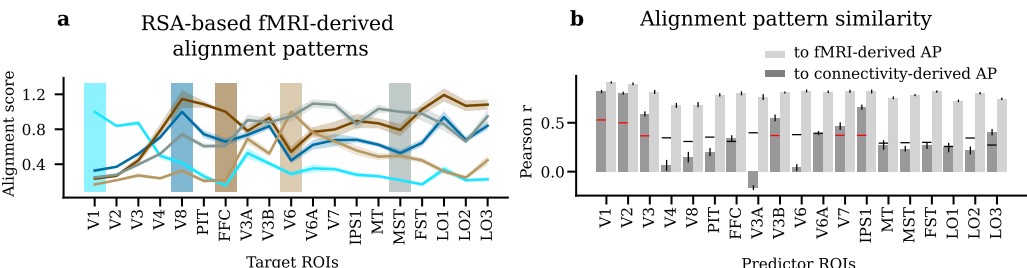

Figure 3: **Brain-brain alignment patterns are consistent across subjects and characteristic for ROIs.** **(a)** RSA-based fMRI-derived AP for example ROIs, mean $\pm$ SEM across subjects. Shaded areas indicate where predictor and target ROI coincide. **(b)** Dark gray bars: APS between fMRI-derived AP and structural connectivity-derived AP, horizontal lines indicate 95% percentile of the null distribution of APS. Red for ROIs where true value is significant, black where it is not, p-values FDR-corrected (see Methods Sec.S2.2).

We found these patterns to be highly consistent across subjects *within* each ROI, and distinct *across* ROIs (Figs. 3a,b; S3.4).

To additionally validate fMRI-derived alignment patterns, we compared them to an independent measure of relations between brain regions: structural connectivity derived from diffusion-weighted imaging in 1065 humans (Caron & Pestilli, 2023, Fig. S3.3). We found high similarity between fMRI-derived and connectivity-derived AP, especially for early regions (Fig. 3b, dark gray bars), and similarity was above chance for ROIs V1, V2, V3, V3B, V7 and IPS1 (see Methods Sec. S2.2). We therefore propose to use fMRI-derived AP as a reference for model-derived AP, extending the NeuroAI Turing Test framework to relational patterns of alignment across brain regions.

### 4.3 ALIGNMENT PATTERNS DIFFERENTIATE BETWEEN MODELS THAT APPEAR EFFECTIVELY EQUIVALENT

We hypothesized that models that achieve similar alignment scores for a given ROI might nonetheless differ in their pattern of alignment scores across ROIs, i.e. their model-derived AP. If this was the case, AP could provide a relational criterion for distinguishing effectively equivalent models. We computed AP for the best-performing models for a given ROI and indeed, we found that equivalently aligned models diverge in their alignment patterns (Fig. 4a; results for LP-based AP Fig. S3.6). Can we say which model is more brain-like in terms of its relational pattern of alignment scores? We calculated alignment pattern similarity between model-derived and fMRI-derived AP (Fig. 4b) and found that (i) all models fell short of brain-brain APS (i.e. the relational NeuroAI Turing Test) and (ii) oftentimes the most brain-aligned models in terms of individual scores were not the most brain-aligned in terms of APS. Especially the high-ranking models from the V-JEPA family achieved low APS with most ROIs (Fig. 4b).

We can apply the same heuristic for effective equivalence on APS, and require that a model, in order to be considered meaningfully aligned to a ROI, should be equivalent to the best-ranking model both on the alignment score and on APS (i.e. it should lie in the region where the gray bars in Fig. 4b intersect). When applying this criterion, the number of candidate models is reduced drastically for most ROIs, and V-JEPA models are ruled out for most ROIs (Fig. 4c).

## 5 DISCUSSION

In this work, we show that standard brain-alignment benchmarking pipelines lack in discriminative power for distinguishing between models that achieve similar alignment scores. We introduce and apply a relational criterion - alignment pattern similarity - and show that it can distinguish between equivalently aligned models. Conceptually, by comparing model-derived alignment patterns

Figure 4: **Alignment patterns differentiate between models that appear effectively equivalent.** **(a)** RSA-based fMRI-derived (solid lines, ROI-color mapping as in Fig. S3.1) and model-derived ((dash-)dotted lines, color mapping as in Fig. 4) AP for four example ROIs. **(b)** Alignment score (RSA) plotted against APS score for all models (gray dots indicate models from the Taskonomy library). Gray shaded areas indicate the 95% CI around the best model's subject-averaged RSA and APS score, respectively. Note that these can be two different models. **(c)** Candidate brain-aligned models (all models effectively equivalent to best performing) without (gray squares, dashed faint lines in a) and with APS requirement (black squares, dash-dotted bold lines in a)). In other words, models that lie within the vertical shaded area in (b) are marked with a gray square; only models that lie in the intersection of the vertical and horizontal shaded areas in (b) are marked with a black square.

to brain-derived alignment patterns, alignment pattern analysis constitutes a relational extension of the NeuroAI Turing Test (Feather et al., 2025).

**Alignment patterns: a tool for distinguishing tools from models.** Consider this intuition: a good model of V1 should not be able to linearly predict higher visual areas, since features grow progressively more complex up the primate visual hierarchy (e.g. Felleman & Essen, 1991; Güçlü & Van Gerven, 2015; Cao & Yamins, 2024a). If an alignment measure (AM) predicts comparable alignment of a model to V1 and to higher visual areas (as is the case e.g. for LP and the V-JEPA models, see Fig. S3.6), this AM lacks the discriminative power to distinguish between models that are good predictive tools for a brain region and models that genuinely resemble its underlying computational mechanisms (Wichmann & Geirhos, 2023). APA operationalizes this intuition and makes it quantifiable. More formally, whereas conventional model-brain alignment benchmarks test similarity in a pointwise fashion (per-ROI alignment), APA adds a second-order structural consistency test. When applied, it exposes the underdetermination of pointwise alignment.

**On the practice of normalization.** This underdetermination is more problematic in regimes of low alignment - in the sense that there are many ways of being poorly aligned to the brain, and only few ways of being strongly aligned. In light of this consideration, both the practice of reporting normalized scores, as well as the question of which reference to normalize against, should be reevaluated. Here, we normalize scores relative to a lower estimate of intersubject consistency calculated from pairwise alignment scores between subjects, following suggestions by Feather et al. (2025). While this procedure increases the discriminative power, especially of LP (compare Figs. 2 and S3.2), it is questionable whether the additional differences that we can detect are really meaningful, given the amount of variance that is left unexplained. An alternative reference is the upper estimate of intersubject consistency calculated between an aggregate reference brain and individual brains (Methods Sec. 3.1.1). Unlike the pairwise reference, this reference is less affected by noise in individual biological measurements. It is substantially higher and may provide a more meaningful target for models.

**The contravariance principle.** These considerations directly connect to the contravariance principle (see e.g. Cao & Yamins, 2024b; Schaeffer et al., 2022). While a given function can be realized through many implementations, increasing task constraints should progressively restrict the space of viable solutions (see also Huh et al., 2024). Task difficulty is one way of increasing such constraints. In principle, this should promote representational convergence and thus stronger brain–model alignment, as has been observed for many years (Kar & DiCarlo, 2024). The recent breakdown of performance–alignment scaling (Linsley et al., 2025), however, suggests that current performance gains are achieved through optimization toward alternative implementations that are not necessarily brain-like. This perspective also suggests that the correlation between task performance and brain alignment in low and intermediate regimes may primarily reflect convergence in generic, task-agnostic representational components rather than task-critical computational strategies. Because natural visual environments exhibit strong statistical regularities, both brains and models may converge toward a shared representational basis adapted towards those statistics, reflected in non-zero alignment scores. However, multiple implementations of task-relevant computations are possible within this shared basis. Consequently, intermediate alignment may largely reflect overlap in general-purpose visual representations rather than convergence toward the brain's task-relevant computational solution Canatar et al. (2023). Alignment pattern analysis can be seen as a way of applying the contravariance principle through additional constraints, independent of task difficulty.

**Towards a stricter framework for brain alignment.** Our findings - together with a growing body of literature (Conwell et al., 2024; Soni et al., 2024; Ahlert et al., 2024; Schaeffer et al., 2024) - suggest that high model-brain alignment scores may be insufficient evidence to draw inferences about model-brain similarity. Progress towards computational models of primate vision with genuine mechanistic similarity requires stricter criteria such as APA, rigorous causal tests (Bowers et al., 2023), and explicit commitments about which scientific claims a given alignment measure is equipped to support - and which not. This becomes all the more relevant as computer vision models grow larger and more powerful: they may become increasingly effective tools for predicting neural responses while at the same time potentially diverging from the brain's computational strategy (Linsley et al., 2025). Without a clearer framework for distinguishing predictive utility from brain likeness, progress on alignment benchmark risks decoupling from scientific insight towards a mechanistic understanding of how the brain works.

ACKNOWLEDGMENTS

We thank all members of the Bethge and Wichmann labs for helpful discussions. KF, MB, and LH were supported by the German Federal Ministry of Research, Technology and Space (PRISMA-V, 01GQ2501B); KF, MB, SK, MT by the German Research Foundation (CRC "Robust Vision", 276693517). KF and LH were additionally supported by the European Research Council (Eye to Action, 101117156). KF was supported by the James Fickel Enigma Project Fund, awarded to Andreas Tolias and Sophia Sanborn.

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

## S1 DISCLOSURE OF LLM USE

We have used LLMs to assist in the code writing process, including for plot creation, to discuss ideas and concepts, in literature search, for searching information in a given work, and for refining text in this paper.

## S2 SUPPLEMENTARY METHODS

### S2.1 ALIGNMENT PATTERN SIMILARITY DISTRIBUTIONS

To assess whether model-brain alignment pattern similarities fall within or outside the distribution of brain-brain alignment pattern similarities (Feather et al., 2025), we first define a subject-specific reference alignment pattern.

For a given ROI $r$ and a subject $t_0$, we compute the mean brain-brain alignment pattern across all pairs of subjects $(p, t)$ in which $t_0$ does not participate, i.e., $p \neq t_0$ and $t \neq t_0$. The subject-specific reference pattern for $p_0$ is then obtained by averaging over all such alignment patterns that exclude $p_0$:

$$\overline{\alpha}_r^{(t_0)} = \frac{1}{|P \setminus \{t_0\}| \cdot |T \setminus \{t_0\}|} \sum_{\substack{p \in P \setminus \{t_0\} \\ t \in T \setminus \{t_0\}}} \alpha_r(\phi_p, \Psi_t), \tag{S1}$$

where $P$ and $T$ denote the sets of all predictor and target subjects.

The brain-brain alignment pattern similarity for subject $t_0$, ROI $r$ is then defined as the average over the similarities between the reference pattern $\overline{\alpha}_r^{(t_0)}$ and all individual alignment patterns **in which** $t_0$ **functions as the target**:

$$\mathbf{D}_{\text{brain}}^{(t_0)} = \left\{ \rho\big(\overline{\alpha}_r^{(t_0)}, \alpha_r(\phi_p, \Psi_{t_0})\big) \ \ \forall p \in P \setminus \{t_0\} \right\}. \tag{S2}$$

Analogously, the model-brain alignment pattern similarity distribution for subject $t_0$ is computed using the model feature space $\phi_m$ as predictor:

$$\mathbf{D}_{\text{model}}^{(t_0)} = \left\{ \rho\big(\overline{\alpha}_{r_0}^{(t_0)}, \alpha_{r_0}(\phi_m, \Psi_{t_0})\big) \right\}. \tag{S3}$$

### S2.2 Significance of fMRI-derived–to–structural alignment pattern similarity

To determine whether an APS value between a structural and an fMRI-derived alignment pattern is meaningful and not due to chance, we create a null distribution of structural patterns, and compute APS between the fMRI-derived pattern to those random patterns.

We create random patterns by sampling 18 regions $\mathbf{k} = (k_1, \ldots, k_{16})$ at random from all regions contained in the full structural connectivity matrix $\tilde{C} = (\tilde{c}_{i,j})_{i,j=1\ldots M}$, $M > N$, containing additional regions to the ones included in our analysis. This yields one random alignment pattern per region,

$$\alpha_{rand,\mathbf{k}}(r) = [\tilde{c}_{\sigma(r),k_1}, \ldots, \tilde{c}_{\sigma(r),k_{16}}] \tag{S4}$$

where $\sigma(r)$ is the index of region $r$ in matrix $\tilde{C}$. We then compute the APS to each subject's fMRI-derived alignment pattern $\alpha(\phi_p^r, \Psi_t)$ as

$$\rho(\alpha(\phi_p^r, \Psi_t), \alpha_{rand,\mathbf{k}}(r))$$

according to equation 3.5.4.

We generated a null distribution by repeating the permutation procedure 1000 times. For each ROI, we computed a group-level test statistic by averaging the correlation values across subjects. The two-sided p-value was calculated as the proportion of absolute correlation values from the null distribution that were greater than or equal to the observed absolute correlation value. To account for multiple comparisons across ROIs, p-values were corrected using the false discovery rate (FDR) procedure Benjamini & Hochberg (1995). Statistical significance was assessed at a threshold of $\alpha = 0.05$ (FDR-corrected).

## S3   Supplementary Results

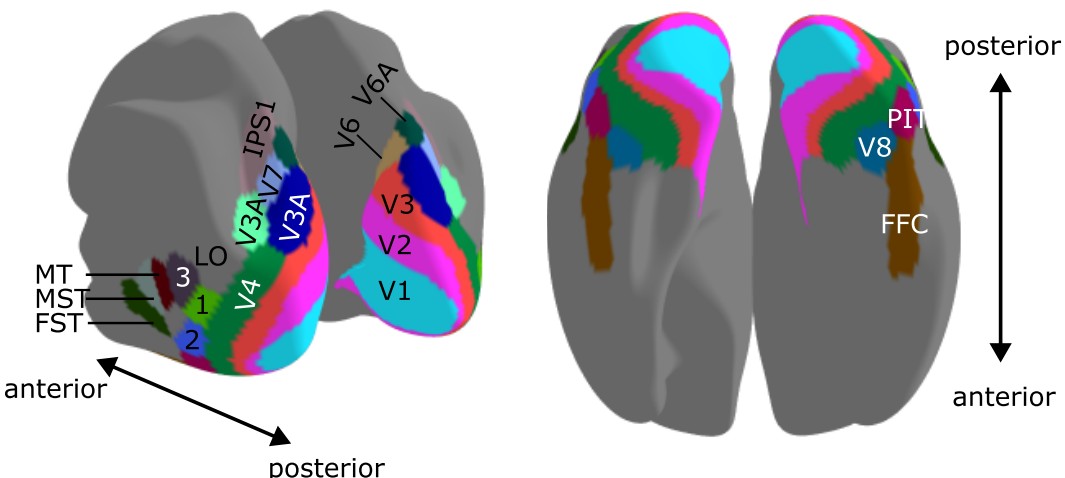

Figure S3.1: Visual ROIs analysed here, mapped onto the HCP inflated cortical surface (both hemispheres combined), defined using the HCP Multi-Modal Parcellation atlas (MMP1.0; Glasser et al., 2016). Lateral view (left) and ventral view (right).

| roi
model | V1 | V2 | V3 | V4 | V8 | PIT | FFC | V3A | V3B | V6 | V6A | V7 | IPS1 | MT | MST | FST | LO1 | LO2 | LO3 |
|---|---|---|---|---|---|---|---|---|---|---|---|---|---|---|---|---|---|---|---|
| egomotion | 0.04 | -0.00 | 0.01 | 0.02 | 0.01 | 0.02 | 0.03 | 0.02 | 0.01 | 0.01 | 0.04 | 0.01 | 0.03 | 0.03 | 0.03 | 0.02 | 0.02 | 0.02 | 0.02 |
| room-layout | 0.06 | 0.01 | 0.03 | 0.02 | 0.01 | 0.02 | 0.03 | 0.01 | 0.00 | -0.00 | 0.04 | 0.01 | 0.02 | 0.04 | 0.03 | 0.02 | 0.03 | 0.02 | 0.02 |
| colorization | 0.09 | 0.02 | 0.04 | 0.03 | 0.01 | 0.01 | 0.02 | 0.01 | 0.00 | -0.00 | 0.02 | 0.00 | 0.02 | 0.03 | 0.03 | 0.02 | 0.04 | 0.02 | 0.02 |
| curvature | 0.06 | 0.01 | 0.02 | 0.02 | 0.00 | 0.02 | 0.03 | 0.01 | 0.01 | -0.00 | 0.04 | 0.02 | 0.03 | 0.04 | 0.03 | 0.02 | 0.03 | 0.02 | 0.02 |
| point-matching | 0.05 | 0.01 | 0.01 | 0.02 | 0.01 | 0.02 | 0.04 | 0.02 | 0.01 | 0.01 | 0.04 | 0.01 | 0.03 | 0.04 | 0.03 | 0.03 | 0.03 | 0.02 | 0.03 |
| keypoints2d | 0.08 | 0.04 | 0.04 | 0.03 | 0.01 | 0.01 | 0.02 | 0.02 | 0.01 | -0.00 | 0.04 | 0.01 | 0.02 | 0.04 | 0.03 | 0.02 | 0.03 | 0.02 | 0.02 |
| fixated-pose | 0.05 | 0.01 | 0.02 | 0.03 | 0.01 | 0.02 | 0.04 | 0.02 | 0.01 | 0.01 | 0.03 | 0.02 | 0.02 | 0.04 | 0.03 | 0.03 | 0.03 | 0.02 | 0.03 |
| edge-texture | 0.12 | 0.03 | 0.04 | 0.04 | 0.01 | 0.01 | 0.01 | 0.01 | 0.01 | -0.01 | 0.04 | 0.01 | 0.02 | 0.03 | 0.03 | 0.02 | 0.02 | 0.02 | 0.01 |
| vanishing-point | 0.06 | 0.02 | 0.03 | 0.03 | 0.02 | 0.03 | 0.03 | 0.02 | -0.00 | -0.00 | 0.03 | 0.02 | 0.02 | 0.04 | 0.03 | 0.02 | 0.04 | 0.03 | 0.03 |
| nonfixated-pose | 0.05 | 0.01 | 0.02 | 0.03 | 0.02 | 0.02 | 0.04 | 0.02 | 0.01 | 0.01 | 0.04 | 0.02 | 0.03 | 0.04 | 0.03 | 0.03 | 0.03 | 0.02 | 0.03 |
| normal | 0.05 | 0.01 | 0.03 | 0.03 | 0.01 | 0.02 | 0.04 | 0.01 | 0.01 | -0.01 | 0.03 | 0.02 | 0.03 | 0.04 | 0.03 | 0.03 | 0.05 | 0.03 | 0.03 |
| segment-semantic | 0.07 | 0.01 | 0.03 | 0.02 | 0.01 | 0.02 | 0.04 | 0.01 | 0.01 | -0.00 | 0.03 | 0.02 | 0.02 | 0.05 | 0.04 | 0.03 | 0.04 | 0.02 | 0.03 |
| edge-occlusion | 0.05 | 0.01 | 0.04 | 0.04 | 0.02 | 0.02 | 0.04 | 0.01 | 0.00 | -0.01 | 0.04 | 0.02 | 0.02 | 0.05 | 0.05 | 0.03 | 0.04 | 0.02 | 0.04 |
| inpainting | 0.07 | 0.04 | 0.04 | 0.04 | 0.03 | 0.02 | 0.04 | 0.03 | 0.01 | 0.02 | 0.03 | 0.01 | 0.03 | 0.03 | 0.03 | 0.02 | 0.03 | 0.02 | 0.04 |
| depth-zbuffer | 0.05 | 0.01 | 0.03 | 0.02 | 0.01 | 0.02 | 0.06 | 0.02 | 0.01 | -0.00 | 0.05 | 0.02 | 0.03 | 0.05 | 0.05 | 0.03 | 0.05 | 0.02 | 0.06 |
| denoising | 0.09 | 0.05 | 0.05 | 0.04 | 0.02 | 0.02 | 0.03 | 0.03 | 0.01 | 0.02 | 0.04 | 0.01 | 0.03 | 0.04 | 0.03 | 0.02 | 0.03 | 0.02 | 0.03 |
| autoencoding | 0.08 | 0.04 | 0.05 | 0.04 | 0.03 | 0.02 | 0.04 | 0.02 | 0.01 | 0.02 | 0.04 | 0.01 | 0.03 | 0.04 | 0.03 | 0.02 | 0.03 | 0.02 | 0.04 |
| segment-unsup25d | 0.06 | 0.03 | 0.05 | 0.03 | 0.02 | 0.02 | 0.04 | 0.03 | 0.01 | 0.01 | 0.05 | 0.02 | 0.03 | 0.06 | 0.05 | 0.03 | 0.04 | 0.03 | 0.04 |
| reshading | 0.06 | 0.01 | 0.04 | 0.03 | 0.02 | 0.03 | 0.06 | 0.02 | 0.01 | -0.00 | 0.04 | 0.02 | 0.03 | 0.06 | 0.05 | 0.04 | 0.05 | 0.04 | 0.04 |
| jigsaw | 0.09 | 0.03 | 0.05 | 0.05 | 0.03 | 0.02 | 0.04 | 0.03 | 0.01 | 0.01 | 0.05 | 0.02 | 0.03 | 0.05 | 0.05 | 0.03 | 0.04 | 0.03 | 0.03 |
| depth-euclidean | 0.06 | 0.02 | 0.04 | 0.04 | 0.03 | 0.04 | 0.06 | 0.03 | 0.01 | -0.00 | 0.05 | 0.02 | 0.04 | 0.06 | 0.06 | 0.04 | 0.05 | 0.04 | 0.04 |
| segment-unsup2d | 0.13 | 0.04 | 0.05 | 0.04 | 0.03 | 0.03 | 0.05 | 0.03 | 0.02 | 0.02 | 0.05 | 0.02 | 0.03 | 0.05 | 0.05 | 0.03 | 0.04 | 0.03 | 0.04 |
| keypoints3d | 0.06 | 0.03 | 0.05 | 0.05 | 0.03 | 0.03 | 0.05 | 0.03 | 0.02 | 0.01 | 0.05 | 0.03 | 0.04 | 0.07 | 0.07 | 0.04 | 0.05 | 0.04 | 0.04 |
| class-scene | 0.06 | 0.03 | 0.05 | 0.06 | 0.05 | 0.04 | 0.07 | 0.04 | 0.02 | 0.01 | 0.06 | 0.03 | 0.04 | 0.08 | 0.08 | 0.06 | 0.06 | 0.04 | 0.06 |
| class-object | 0.06 | 0.04 | 0.06 | 0.06 | 0.04 | 0.04 | 0.07 | 0.03 | 0.02 | 0.01 | 0.06 | 0.03 | 0.04 | 0.08 | 0.07 | 0.05 | 0.06 | 0.05 | 0.06 |
| timm/convnext-large | 0.12 | 0.11 | 0.13 | 0.13 | 0.05 | 0.05 | 0.07 | 0.07 | 0.06 | 0.02 | 0.04 | 0.02 | 0.04 | 0.06 | 0.05 | 0.05 | 0.04 | 0.04 | 0.05 |
| clip/ViT-L/14@336px | 0.12 | 0.11 | 0.11 | 0.09 | 0.05 | 0.07 | 0.09 | 0.05 | 0.04 | 0.02 | 0.07 | 0.04 | 0.06 | 0.10 | 0.09 | 0.07 | 0.08 | 0.08 | 0.07 |
| timm/convnext-base | 0.12 | 0.11 | 0.13 | 0.13 | 0.05 | 0.05 | 0.10 | 0.05 | 0.05 | 0.03 | 0.06 | 0.04 | 0.06 | 0.09 | 0.08 | 0.07 | 0.07 | 0.07 | 0.08 |
| clip/ViT-L/14 | 0.11 | 0.08 | 0.10 | 0.11 | 0.06 | 0.08 | 0.10 | 0.05 | 0.04 | 0.02 | 0.09 | 0.04 | 0.08 | 0.10 | 0.09 | 0.08 | 0.08 | 0.08 | 0.07 |
| timm/convnext-small | 0.10 | 0.09 | 0.11 | 0.10 | 0.07 | 0.08 | 0.12 | 0.06 | 0.05 | 0.04 | 0.08 | 0.05 | 0.06 | 0.10 | 0.11 | 0.09 | 0.08 | 0.09 | 0.09 |
| clip/ViT-B/16 | 0.13 | 0.10 | 0.11 | 0.12 | 0.07 | 0.08 | 0.11 | 0.06 | 0.05 | 0.03 | 0.08 | 0.05 | 0.07 | 0.10 | 0.10 | 0.08 | 0.09 | 0.09 | 0.08 |
| v-jepa/vjepa2-vit-huge | 0.11 | 0.13 | 0.12 | 0.12 | 0.09 | 0.07 | 0.10 | 0.08 | 0.06 | 0.08 | 0.08 | 0.05 | 0.05 | 0.09 | 0.10 | 0.08 | 0.07 | 0.08 | 0.08 |
| opt-cwm/opt-cwm | 0.09 | 0.11 | 0.11 | 0.09 | 0.05 | 0.07 | 0.08 | 0.09 | 0.06 | 0.17 | 0.10 | 0.07 | 0.06 | 0.10 | 0.13 | 0.09 | 0.06 | 0.07 | 0.07 |
| mmaction2/swin-tiny | 0.13 | 0.11 | 0.13 | 0.11 | 0.09 | 0.08 | 0.12 | 0.08 | 0.06 | 0.05 | 0.08 | 0.06 | 0.07 | 0.10 | 0.11 | 0.10 | 0.09 | 0.09 | 0.10 |
| timm/convnext-tiny | 0.07 | 0.04 | 0.06 | 0.11 | 0.09 | 0.10 | 0.14 | 0.07 | 0.06 | 0.05 | 0.10 | 0.07 | 0.08 | 0.14 | 0.14 | 0.12 | 0.10 | 0.11 | 0.11 |
| mmaction2/mvit-small | 0.21 | 0.16 | 0.14 | 0.12 | 0.09 | 0.07 | 0.10 | 0.09 | 0.06 | 0.07 | 0.08 | 0.06 | 0.05 | 0.10 | 0.10 | 0.08 | 0.07 | 0.08 | 0.08 |
| mmaction2/timesformer | 0.15 | 0.14 | 0.13 | 0.13 | 0.09 | 0.09 | 0.13 | 0.07 | 0.05 | 0.06 | 0.08 | 0.06 | 0.06 | 0.11 | 0.11 | 0.09 | 0.09 | 0.10 | 0.10 |
| timm/resnet18 | 0.12 | 0.10 | 0.11 | 0.15 | 0.11 | 0.11 | 0.18 | 0.09 | 0.07 | 0.06 | 0.11 | 0.08 | 0.09 | 0.16 | 0.15 | 0.14 | 0.12 | 0.13 | 0.14 |
| vissl/simclr-resnet101 | 0.12 | 0.09 | 0.13 | 0.18 | 0.14 | 0.15 | 0.22 | 0.09 | 0.07 | 0.05 | 0.10 | 0.08 | 0.09 | 0.17 | 0.16 | 0.15 | 0.15 | 0.16 | 0.17 |
| clip/ViT-B/32 | 0.13 | 0.11 | 0.13 | 0.18 | 0.14 | 0.15 | 0.23 | 0.10 | 0.09 | 0.06 | 0.12 | 0.09 | 0.11 | 0.19 | 0.18 | 0.16 | 0.16 | 0.17 | 0.17 |
| vissl/simclr-resnet50 | 0.13 | 0.11 | 0.14 | 0.21 | 0.15 | 0.17 | 0.23 | 0.09 | 0.08 | 0.06 | 0.10 | 0.08 | 0.10 | 0.19 | 0.17 | 0.17 | 0.16 | 0.18 | 0.18 |
| timm/resnet50 | 0.12 | 0.11 | 0.13 | 0.19 | 0.16 | 0.16 | 0.24 | 0.10 | 0.09 | 0.08 | 0.12 | 0.09 | 0.10 | 0.19 | 0.18 | **0.17** | 0.15 | 0.17 | 0.19 |
| vggt/VGGT-1B | 0.14 | 0.09 | 0.11 | 0.20 | 0.15 | 0.15 | 0.24 | 0.10 | 0.10 | 0.08 | 0.13 | 0.09 | **0.12** | **0.20** | 0.20 | 0.16 | 0.15 | 0.17 | **0.20** |
| clip/RN50 | 0.13 | 0.11 | 0.13 | 0.20 | 0.15 | **0.17** | **0.25** | 0.10 | 0.10 | 0.06 | 0.12 | 0.09 | 0.11 | 0.20 | 0.19 | 0.17 | **0.17** | **0.18** | 0.19 |
| v-jepa/vjepa2-vit-giant | 0.17 | 0.19 | 0.19 | 0.23 | **0.17** | 0.13 | 0.16 | 0.15 | 0.12 | 0.15 | 0.14 | 0.11 | 0.09 | 0.15 | 0.18 | 0.15 | 0.14 | 0.14 | 0.17 |
| v-jepa/vjepa2-vit-large | 0.17 | 0.22 | 0.20 | 0.24 | 0.17 | 0.13 | 0.18 | 0.16 | 0.13 | 0.15 | 0.13 | 0.12 | 0.10 | 0.17 | 0.19 | 0.15 | 0.13 | 0.14 | 0.16 |
| v-jepa/vjepa2-vit-giant-384 | **0.22** | **0.28** | **0.24** | **0.25** | 0.17 | 0.13 | 0.17 | **0.19** | **0.14** | **0.20** | **0.15** | **0.12** | 0.11 | 0.18 | **0.22** | 0.14 | 0.13 | 0.13 | 0.16 |

Table S3.1: Model-brain alignment under RSA.

| roi
model | V1 | V2 | V3 | V4 | V8 | PIT | FFC | V3A | V3B | V6 | V6A | V7 | IPS1 | MT | MST | FST | LO1 | LO2 | LO3 |
|---|---|---|---|---|---|---|---|---|---|---|---|---|---|---|---|---|---|---|---|
| egomotion | 0.12 | -0.00 | 0.05 | 0.06 | 0.03 | 0.04 | 0.05 | 0.04 | 0.04 | 0.04 | 0.14 | 0.07 | 0.12 | 0.10 | 0.08 | 0.06 | 0.09 | 0.04 | 0.07 |
| room-layout | 0.17 | 0.03 | 0.08 | 0.06 | 0.02 | 0.05 | 0.05 | 0.02 | 0.03 | -0.02 | 0.13 | 0.06 | 0.10 | 0.10 | 0.08 | 0.05 | 0.12 | 0.06 | 0.06 |
| colorization | 0.24 | 0.06 | 0.11 | 0.07 | 0.05 | 0.03 | 0.04 | -0.01 | 0.02 | -0.02 | 0.06 | 0.02 | 0.07 | 0.10 | 0.07 | 0.04 | 0.13 | 0.05 | 0.06 |
| curvature | 0.15 | 0.02 | 0.06 | 0.05 | 0.02 | 0.06 | 0.05 | 0.02 | 0.04 | -0.02 | 0.13 | 0.08 | 0.14 | 0.11 | 0.09 | 0.05 | 0.13 | 0.06 | 0.07 |
| point-matching | 0.13 | 0.02 | 0.07 | 0.06 | 0.04 | 0.05 | 0.06 | 0.06 | 0.05 | 0.05 | 0.14 | 0.07 | 0.15 | 0.11 | 0.09 | 0.07 | 0.10 | 0.05 | 0.08 |
| keypoints2d | 0.23 | 0.09 | 0.13 | 0.07 | 0.04 | 0.03 | 0.03 | 0.04 | 0.04 | -0.01 | 0.14 | 0.05 | 0.09 | 0.10 | 0.07 | 0.05 | 0.11 | 0.06 | 0.05 |
| fixated-pose | 0.14 | 0.02 | 0.07 | 0.07 | 0.05 | 0.05 | 0.06 | 0.06 | 0.06 | 0.04 | 0.09 | 0.10 | 0.10 | 0.11 | 0.09 | 0.07 | 0.12 | 0.05 | 0.09 |
| edge-texture | 0.34 | 0.09 | 0.12 | 0.09 | 0.04 | 0.03 | 0.01 | -0.00 | 0.04 | -0.02 | 0.14 | 0.07 | 0.09 | 0.09 | 0.08 | 0.04 | 0.09 | 0.06 | 0.03 |
| vanishing-point | 0.16 | 0.05 | 0.09 | 0.07 | 0.05 | 0.07 | 0.06 | 0.04 | -0.00 | -0.02 | 0.11 | 0.07 | 0.08 | 0.11 | 0.09 | 0.06 | 0.15 | 0.08 | 0.08 |
| nonfixated-pose | 0.13 | 0.02 | 0.07 | 0.07 | 0.07 | 0.05 | 0.07 | 0.06 | 0.06 | 0.04 | 0.15 | 0.08 | 0.14 | 0.11 | 0.08 | 0.08 | 0.10 | 0.05 | 0.10 |
| normal | 0.14 | 0.03 | 0.10 | 0.06 | 0.04 | 0.06 | 0.07 | 0.03 | 0.04 | -0.06 | 0.12 | 0.08 | 0.13 | 0.12 | 0.09 | 0.07 | 0.17 | 0.08 | 0.09 |
| segment-semantic | 0.18 | 0.02 | 0.08 | 0.05 | 0.03 | 0.04 | 0.06 | 0.02 | 0.06 | 0.00 | 0.12 | 0.09 | 0.11 | 0.16 | 0.12 | 0.08 | 0.15 | 0.06 | 0.09 |
| edge-occlusion | 0.15 | 0.03 | 0.12 | 0.09 | 0.07 | 0.05 | 0.08 | 0.03 | 0.03 | -0.03 | 0.13 | 0.08 | 0.11 | 0.16 | 0.12 | 0.08 | 0.14 | 0.06 | 0.11 |
| inpainting | 0.20 | 0.09 | 0.12 | 0.09 | 0.09 | 0.05 | 0.06 | 0.08 | 0.06 | 0.10 | 0.15 | 0.06 | 0.11 | 0.10 | 0.07 | 0.06 | 0.11 | 0.05 | 0.10 |
| depth-zbuffer | 0.15 | 0.03 | 0.09 | 0.06 | 0.05 | 0.06 | 0.10 | 0.05 | 0.05 | -0.01 | 0.18 | 0.08 | 0.15 | 0.15 | 0.11 | 0.06 | 0.17 | 0.07 | 0.18 |
| denoising | 0.25 | 0.13 | 0.15 | 0.09 | 0.08 | 0.05 | 0.06 | 0.09 | 0.07 | 0.09 | 0.15 | 0.06 | 0.13 | 0.11 | 0.08 | 0.05 | 0.13 | 0.06 | 0.10 |
| autoencoding | 0.22 | 0.10 | 0.14 | 0.10 | 0.10 | 0.05 | 0.07 | 0.07 | 0.07 | 0.10 | 0.15 | 0.06 | 0.13 | 0.11 | 0.08 | 0.06 | 0.13 | 0.06 | 0.11 |
| segment-unsup25d | 0.16 | 0.08 | 0.14 | 0.08 | 0.08 | 0.06 | 0.07 | 0.10 | 0.05 | 0.04 | 0.20 | 0.12 | 0.13 | 0.16 | 0.13 | 0.09 | 0.17 | 0.07 | 0.11 |
| reshading | 0.17 | 0.04 | 0.10 | 0.08 | 0.07 | 0.07 | 0.10 | 0.06 | 0.07 | -0.01 | 0.16 | 0.11 | 0.15 | 0.17 | 0.13 | 0.12 | 0.19 | 0.10 | 0.13 |
| jigsaw | 0.26 | 0.08 | 0.14 | 0.13 | 0.09 | 0.06 | 0.07 | 0.07 | 0.08 | 0.05 | 0.18 | 0.11 | 0.12 | 0.14 | 0.11 | 0.07 | 0.14 | 0.07 | 0.10 |
| depth-euclidean | 0.15 | 0.06 | 0.12 | 0.09 | 0.10 | 0.11 | 0.11 | 0.09 | 0.07 | 0.01 | 0.18 | 0.12 | 0.17 | 0.17 | 0.14 | 0.12 | 0.20 | 0.10 | 0.13 |
| segment-unsup2d | 0.34 | 0.11 | 0.15 | 0.12 | 0.11 | 0.07 | 0.08 | 0.09 | 0.10 | 0.09 | 0.19 | 0.09 | 0.13 | 0.15 | 0.14 | 0.08 | 0.16 | 0.09 | 0.12 |
| keypoints3d | 0.16 | 0.08 | 0.15 | 0.10 | 0.10 | 0.08 | 0.09 | 0.09 | 0.10 | 0.05 | 0.21 | 0.16 | 0.17 | 0.19 | 0.18 | 0.12 | 0.21 | 0.11 | 0.13 |
| class-scene | 0.17 | 0.08 | 0.15 | 0.12 | 0.17 | 0.10 | 0.13 | 0.12 | 0.10 | 0.05 | 0.21 | 0.14 | 0.18 | 0.23 | 0.19 | 0.16 | 0.22 | 0.12 | 0.18 |
| class-object | 0.16 | 0.10 | 0.18 | 0.13 | 0.15 | 0.10 | 0.12 | 0.11 | 0.11 | 0.03 | 0.23 | 0.16 | 0.19 | 0.22 | 0.17 | 0.13 | 0.23 | 0.13 | 0.17 |
| timm/convnext-large | 0.31 | 0.26 | 0.36 | 0.27 | 0.19 | 0.12 | 0.13 | 0.28 | 0.25 | 0.11 | 0.15 | 0.11 | 0.19 | 0.18 | 0.12 | 0.13 | 0.16 | 0.12 | 0.15 |
| clip/ViT-L/14@336px | 0.30 | 0.25 | 0.30 | 0.20 | 0.18 | 0.18 | 0.16 | 0.18 | 0.21 | 0.08 | 0.30 | 0.22 | 0.29 | 0.28 | 0.22 | 0.20 | 0.32 | 0.20 | 0.21 |
| timm/convnext-base | 0.31 | 0.26 | 0.36 | 0.28 | 0.17 | 0.18 | 0.19 | 0.19 | 0.20 | 0.15 | 0.23 | 0.18 | 0.27 | 0.25 | 0.19 | 0.20 | 0.25 | 0.20 | 0.22 |
| clip/ViT-L/14 | 0.28 | 0.19 | 0.27 | 0.23 | 0.19 | 0.21 | 0.17 | 0.20 | 0.19 | 0.09 | 0.34 | 0.22 | 0.39 | 0.27 | 0.21 | 0.21 | 0.30 | 0.22 | 0.21 |
| timm/convnext-small | 0.27 | 0.22 | 0.30 | 0.21 | 0.22 | 0.20 | 0.21 | 0.26 | 0.24 | 0.22 | 0.31 | 0.25 | 0.29 | 0.30 | 0.25 | 0.26 | 0.29 | 0.22 | 0.27 |
| clip/ViT-B/16 | 0.34 | 0.23 | 0.31 | 0.26 | 0.24 | 0.22 | 0.20 | 0.24 | 0.25 | 0.12 | 0.33 | 0.25 | 0.33 | 0.29 | 0.24 | 0.23 | 0.33 | 0.22 | 0.25 |
| v-jepa/vjepa2-vit-huge | 0.28 | 0.31 | 0.33 | 0.25 | 0.30 | 0.18 | 0.18 | 0.35 | 0.28 | 0.35 | 0.32 | 0.25 | 0.23 | 0.25 | 0.24 | 0.23 | 0.29 | 0.21 | 0.24 |
| opt-cwm/opt-cwm | 0.24 | 0.27 | 0.30 | 0.17 | 0.17 | 0.17 | 0.15 | 0.40 | 0.26 | 0.70 | 0.41 | 0.36 | 0.28 | 0.29 | 0.29 | 0.24 | 0.24 | 0.17 | 0.21 |
| mmaction2/swin-tiny | 0.35 | 0.25 | 0.35 | 0.23 | 0.30 | 0.20 | 0.22 | 0.32 | 0.27 | 0.22 | 0.34 | 0.29 | 0.31 | 0.30 | 0.26 | 0.27 | 0.33 | 0.23 | 0.29 |
| timm/convnext-tiny | 0.19 | 0.10 | 0.16 | 0.24 | 0.29 | 0.26 | 0.26 | 0.28 | 0.28 | 0.24 | 0.41 | 0.32 | 0.39 | 0.39 | 0.34 | 0.34 | 0.37 | 0.29 | 0.34 |
| mmaction2/mvit-small | 0.56 | 0.37 | 0.40 | 0.26 | 0.31 | 0.19 | 0.19 | 0.39 | 0.29 | 0.30 | 0.34 | 0.29 | 0.23 | 0.28 | 0.23 | 0.22 | 0.28 | 0.20 | 0.24 |
| mmaction2/timesformer | 0.39 | 0.33 | 0.37 | 0.28 | 0.31 | 0.24 | 0.23 | 0.31 | 0.23 | 0.23 | 0.32 | 0.29 | 0.28 | 0.30 | 0.25 | 0.25 | 0.35 | 0.27 | 0.30 |
| timm/resnet18 | 0.33 | 0.24 | 0.32 | 0.32 | 0.38 | 0.33 | 0.32 | 0.36 | 0.31 | 0.24 | 0.45 | 0.38 | 0.45 | 0.45 | 0.36 | 0.40 | 0.44 | 0.35 | 0.42 |
| vissl/simclr-resnet101 | 0.32 | 0.22 | 0.35 | 0.39 | 0.45 | 0.40 | 0.39 | 0.35 | 0.33 | 0.25 | 0.42 | 0.39 | 0.43 | 0.48 | 0.38 | 0.43 | 0.57 | 0.42 | 0.48 |
| clip/ViT-B/32 | 0.34 | 0.26 | 0.37 | 0.39 | 0.48 | 0.41 | 0.41 | 0.40 | 0.42 | 0.27 | 0.50 | 0.43 | 0.51 | 0.53 | 0.43 | 0.45 | 0.59 | 0.44 | 0.51 |
| vissl/simclr-resnet50 | 0.35 | 0.27 | 0.40 | 0.44 | 0.51 | 0.44 | 0.42 | 0.38 | 0.38 | 0.28 | 0.43 | 0.41 | 0.46 | 0.52 | 0.41 | 0.46 | 0.61 | **0.47** | 0.52 |
| timm/resnet50 | 0.33 | 0.25 | 0.36 | 0.42 | 0.52 | 0.41 | 0.42 | 0.41 | 0.42 | 0.36 | 0.50 | 0.46 | 0.48 | 0.53 | 0.43 | **0.48** | 0.56 | 0.45 | 0.55 |
| vggt/VGGT-1B | 0.39 | 0.23 | 0.32 | 0.42 | 0.49 | 0.39 | 0.43 | 0.42 | 0.46 | 0.35 | 0.55 | 0.43 | **0.59** | **0.57** | 0.45 | 0.45 | 0.57 | 0.43 | **0.58** |
| clip/RN50 | 0.33 | 0.26 | 0.37 | 0.43 | 0.52 | **0.46** | **0.45** | 0.41 | 0.44 | 0.28 | 0.49 | 0.44 | 0.55 | 0.55 | 0.45 | 0.47 | **0.62** | 0.47 | 0.54 |
| v-jepa/vjepa2-vit-giant | 0.46 | 0.45 | 0.51 | 0.49 | **0.57** | 0.33 | 0.29 | 0.64 | 0.52 | 0.65 | 0.58 | 0.53 | 0.43 | 0.48 | 0.42 | 0.42 | 0.53 | 0.37 | 0.51 |
| v-jepa/vjepa2-vit-large | 0.44 | 0.51 | 0.56 | 0.51 | 0.57 | 0.35 | 0.33 | 0.71 | 0.56 | 0.66 | 0.58 | 0.59 | 0.48 | 0.48 | 0.45 | 0.43 | 0.50 | 0.38 | 0.49 |
| v-jepa/vjepa2-vit-giant-384 | **0.57** | **0.64** | **0.68** | **0.53** | 0.56 | 0.34 | 0.30 | **0.85** | **0.61** | **0.86** | **0.62** | **0.61** | 0.50 | 0.51 | **0.51** | 0.41 | 0.51 | 0.34 | 0.47 |

Table S3.2: Model-brain alignment under RSA, normalized to NeuroAI Turing Test.

| roi model | V1 | V2 | V3 | V4 | V8 | PIT | FFC | V3A | V3B | V6 | V6A | V7 | IPS1 | MT | MST | FST | LO1 | LO2 | LO3 |
|---|---|---|---|---|---|---|---|---|---|---|---|---|---|---|---|---|---|---|---|
| egomotion | 0.03 | 0.01 | 0.01 | 0.00 | 0.02 | 0.03 | 0.03 | -0.02 | -0.01 | 0.01 | 0.01 | -0.00 | 0.01 | 0.04 | 0.04 | 0.03 | 0.01 | 0.03 | 0.02 |
| room-layout | 0.06 | 0.02 | 0.02 | 0.02 | 0.03 | 0.05 | 0.05 | -0.00 | 0.01 | 0.01 | 0.02 | 0.02 | 0.02 | 0.04 | 0.04 | 0.03 | 0.05 | 0.05 | 0.05 |
| colorization | 0.07 | 0.05 | 0.05 | 0.04 | 0.05 | 0.07 | 0.04 | -0.00 | -0.00 | 0.01 | 0.01 | -0.00 | 0.01 | 0.05 | 0.04 | 0.05 | 0.05 | 0.07 | 0.04 |
| curvature | 0.06 | 0.04 | 0.03 | 0.02 | 0.03 | 0.04 | 0.04 | -0.00 | 0.00 | 0.02 | 0.02 | 0.00 | 0.01 | 0.04 | 0.05 | 0.03 | 0.03 | 0.05 | 0.03 |
| point-matching | 0.04 | 0.02 | 0.02 | 0.01 | 0.02 | 0.02 | 0.03 | -0.01 | 0.00 | 0.02 | 0.01 | 0.01 | 0.02 | 0.05 | 0.04 | 0.03 | 0.01 | 0.03 | 0.03 |
| keypoints2d | 0.04 | 0.01 | 0.01 | -0.00 | -0.00 | 0.02 | 0.01 | -0.01 | -0.01 | 0.00 | -0.00 | -0.01 | 0.00 | 0.02 | 0.02 | 0.01 | 0.01 | 0.03 | 0.01 |
| fixated-pose | 0.03 | 0.01 | 0.00 | 0.00 | 0.02 | 0.03 | 0.04 | -0.01 | -0.00 | 0.01 | 0.01 | 0.01 | 0.02 | 0.02 | 0.02 | 0.02 | 0.02 | 0.04 | 0.03 |
| edge-texture | 0.08 | 0.04 | 0.05 | 0.04 | 0.03 | 0.04 | 0.03 | -0.01 | -0.00 | 0.01 | 0.01 | -0.01 | 0.01 | 0.03 | 0.03 | 0.02 | 0.03 | 0.05 | 0.03 |
| vanishing-point | 0.05 | 0.02 | 0.02 | 0.02 | 0.02 | 0.04 | 0.04 | -0.00 | -0.01 | 0.02 | -0.00 | -0.01 | 0.00 | 0.05 | 0.05 | 0.05 | 0.04 | 0.05 | 0.04 |
| nonfixated-pose | 0.02 | -0.00 | -0.01 | -0.01 | -0.00 | 0.01 | 0.02 | -0.01 | -0.00 | 0.00 | 0.01 | -0.00 | 0.01 | 0.03 | 0.02 | 0.01 | -0.00 | 0.02 | 0.01 |
| normal | 0.06 | 0.04 | 0.06 | 0.06 | 0.06 | 0.09 | 0.07 | 0.01 | 0.02 | 0.03 | 0.03 | 0.03 | 0.03 | 0.09 | 0.07 | 0.06 | 0.08 | 0.09 | 0.08 |
| segment-semantic | 0.08 | 0.05 | 0.06 | 0.07 | 0.06 | 0.09 | 0.07 | 0.02 | 0.03 | 0.03 | 0.04 | 0.03 | 0.04 | 0.08 | 0.07 | 0.06 | 0.08 | 0.10 | 0.07 |
| edge-occlusion | 0.08 | 0.05 | 0.06 | 0.06 | 0.07 | 0.08 | 0.08 | 0.01 | 0.02 | 0.03 | 0.03 | 0.01 | 0.03 | 0.10 | 0.08 | 0.05 | 0.07 | 0.10 | 0.07 |
| inpainting | 0.03 | 0.01 | -0.00 | -0.00 | 0.01 | 0.04 | 0.04 | -0.01 | -0.01 | 0.01 | 0.00 | -0.01 | 0.00 | 0.03 | 0.04 | 0.02 | 0.02 | 0.05 | 0.02 |
| depth-zbuffer | 0.07 | 0.04 | 0.05 | 0.06 | 0.06 | 0.09 | 0.06 | 0.01 | 0.03 | 0.03 | 0.03 | 0.02 | 0.04 | 0.08 | 0.07 | 0.05 | 0.07 | 0.10 | 0.07 |
| denoising | 0.02 | 0.01 | -0.00 | -0.02 | -0.01 | 0.02 | 0.01 | -0.01 | -0.01 | 0.01 | 0.00 | -0.01 | 0.00 | 0.01 | 0.02 | 0.01 | 0.00 | 0.02 | 0.01 |
| autoencoding | 0.01 | -0.00 | -0.01 | -0.01 | -0.01 | 0.01 | 0.01 | -0.01 | -0.01 | 0.01 | 0.00 | -0.01 | 0.00 | 0.01 | 0.02 | 0.01 | 0.00 | 0.02 | 0.00 |
| segment-unsup25d | 0.06 | 0.04 | 0.06 | 0.06 | 0.07 | 0.09 | 0.07 | 0.01 | 0.03 | 0.02 | 0.03 | 0.02 | 0.03 | 0.07 | 0.06 | 0.05 | 0.07 | 0.10 | 0.07 |
| reshading | 0.06 | 0.04 | 0.07 | 0.07 | 0.08 | 0.11 | 0.10 | 0.01 | 0.04 | 0.04 | 0.04 | 0.03 | 0.04 | 0.10 | 0.08 | 0.07 | 0.09 | 0.13 | 0.09 |
| jigsaw | 0.05 | 0.02 | 0.02 | 0.02 | 0.03 | 0.05 | 0.05 | -0.01 | 0.01 | 0.03 | 0.02 | 0.00 | 0.03 | 0.05 | 0.05 | 0.03 | 0.03 | 0.05 | 0.04 |
| depth-euclidean | 0.06 | 0.05 | 0.06 | 0.07 | 0.08 | 0.10 | 0.08 | 0.02 | 0.03 | 0.03 | 0.03 | 0.02 | 0.04 | 0.09 | 0.08 | 0.06 | 0.08 | 0.10 | 0.09 |
| segment-unsup2d | 0.09 | 0.05 | 0.05 | 0.04 | 0.04 | 0.06 | 0.05 | -0.00 | -0.00 | 0.02 | 0.01 | -0.01 | 0.02 | 0.05 | 0.04 | 0.03 | 0.04 | 0.06 | 0.03 |
| keypoints3d | 0.08 | 0.05 | 0.06 | 0.08 | 0.07 | 0.11 | 0.09 | 0.01 | 0.03 | 0.03 | 0.03 | 0.03 | 0.04 | 0.10 | 0.09 | 0.06 | 0.08 | 0.12 | 0.08 |
| class-scene | 0.09 | 0.07 | 0.08 | 0.09 | 0.11 | 0.11 | 0.10 | 0.03 | 0.06 | 0.04 | 0.06 | 0.06 | 0.06 | 0.11 | 0.09 | 0.08 | 0.11 | 0.11 | 0.10 |
| class-object | 0.10 | 0.09 | 0.09 | 0.11 | 0.12 | 0.13 | 0.12 | 0.04 | 0.07 | 0.04 | 0.06 | 0.06 | 0.07 | 0.13 | 0.10 | 0.10 | 0.12 | 0.14 | 0.13 |
| timm/convnext-large | 0.14 | 0.12 | 0.14 | 0.19 | 0.19 | 0.23 | 0.21 | 0.08 | 0.14 | 0.06 | 0.11 | 0.12 | 0.12 | 0.26 | 0.20 | 0.20 | 0.23 | 0.27 | 0.24 |
| clip/ViT-L/14@336px | 0.15 | 0.13 | 0.14 | 0.19 | 0.19 | 0.24 | 0.22 | 0.07 | 0.14 | 0.08 | 0.12 | 0.13 | 0.14 | 0.28 | 0.24 | 0.23 | 0.23 | 0.27 | 0.25 |
| timm/convnext-base | 0.13 | 0.12 | 0.15 | 0.20 | 0.21 | 0.25 | 0.22 | 0.09 | 0.15 | 0.09 | 0.11 | 0.13 | 0.14 | 0.24 | 0.22 | 0.21 | 0.24 | 0.28 | 0.22 |
| clip/ViT-L/14 | 0.14 | 0.12 | 0.14 | 0.19 | 0.19 | 0.23 | 0.23 | 0.07 | 0.15 | 0.09 | 0.12 | 0.12 | 0.14 | 0.28 | 0.24 | 0.24 | 0.22 | 0.28 | 0.24 |
| timm/convnext-small | 0.12 | 0.13 | 0.15 | 0.20 | 0.18 | 0.25 | 0.21 | 0.07 | 0.13 | 0.07 | 0.11 | 0.13 | 0.12 | 0.28 | 0.25 | 0.22 | 0.24 | 0.28 | 0.23 |
| clip/ViT-B/16 | 0.14 | 0.11 | 0.15 | 0.18 | 0.19 | 0.24 | 0.22 | 0.07 | 0.15 | 0.08 | 0.12 | 0.15 | 0.14 | 0.27 | 0.23 | 0.23 | 0.23 | 0.28 | 0.23 |
| v-jepa/vjepa2-vit-huge | 0.13 | 0.15 | 0.18 | 0.22 | 0.22 | 0.25 | 0.20 | 0.13 | 0.18 | 0.14 | 0.18 | 0.20 | 0.13 | 0.35 | 0.35 | 0.26 | 0.26 | 0.29 | 0.26 |
| opt-cwm/opt-cwm | 0.15 | 0.14 | 0.16 | 0.19 | 0.17 | 0.20 | 0.16 | 0.12 | 0.15 | 0.13 | 0.15 | 0.16 | 0.11 | 0.25 | 0.24 | 0.19 | 0.19 | 0.23 | 0.18 |
| mmaction2/swin-tiny | 0.11 | 0.10 | 0.12 | 0.14 | 0.16 | 0.19 | 0.17 | 0.07 | 0.10 | 0.07 | 0.09 | 0.09 | 0.10 | 0.26 | 0.22 | 0.21 | 0.18 | 0.22 | 0.20 |
| timm/convnext-tiny | 0.13 | 0.12 | 0.15 | 0.17 | 0.17 | 0.22 | 0.20 | 0.07 | 0.12 | 0.08 | 0.10 | 0.10 | 0.12 | 0.27 | 0.23 | 0.21 | 0.22 | 0.26 | 0.24 |
| mmaction2/mvit-small | 0.16 | 0.14 | 0.15 | 0.17 | 0.17 | 0.22 | 0.21 | 0.08 | 0.11 | 0.10 | 0.10 | 0.10 | 0.12 | 0.28 | 0.25 | 0.21 | 0.16 | 0.24 | 0.21 |
| mmaction2/timesformer | 0.07 | 0.05 | 0.06 | 0.07 | 0.09 | 0.11 | 0.10 | 0.03 | 0.05 | 0.04 | 0.05 | 0.05 | 0.05 | 0.11 | 0.09 | 0.10 | 0.10 | 0.12 | 0.11 |
| timm/resnet18 | 0.13 | 0.11 | 0.14 | 0.18 | 0.20 | 0.26 | 0.23 | 0.08 | 0.15 | 0.09 | 0.12 | 0.15 | 0.14 | 0.30 | 0.25 | 0.25 | 0.25 | 0.27 | 0.25 |
| vissl/simclr-resnet101 | 0.15 | 0.13 | 0.16 | 0.21 | 0.22 | 0.27 | 0.25 | 0.10 | 0.16 | 0.09 | 0.14 | 0.16 | 0.16 | 0.30 | 0.26 | 0.24 | 0.26 | 0.30 | 0.26 |
| clip/ViT-B/32 | 0.15 | 0.13 | 0.16 | 0.22 | 0.23 | 0.29 | 0.27 | 0.10 | 0.17 | 0.11 | 0.15 | 0.17 | 0.17 | 0.34 | 0.28 | 0.27 | 0.29 | 0.33 | 0.30 |
| vissl/simclr-resnet50 | 0.15 | 0.12 | 0.15 | 0.21 | 0.22 | 0.28 | 0.25 | 0.10 | 0.16 | 0.10 | 0.13 | 0.14 | 0.15 | 0.30 | 0.25 | 0.23 | 0.25 | 0.31 | 0.27 |
| timm/resnet50 | 0.12 | 0.11 | 0.13 | 0.19 | 0.21 | 0.28 | 0.25 | 0.08 | 0.16 | 0.10 | 0.13 | 0.16 | 0.16 | 0.33 | 0.28 | 0.26 | 0.26 | 0.31 | 0.28 |
| vggt/VGGT-1B | 0.16 | 0.14 | 0.16 | 0.22 | 0.23 | 0.31 | 0.29 | 0.12 | 0.20 | 0.12 | 0.18 | 0.20 | **0.19** | 0.35 | 0.31 | 0.28 | 0.30 | 0.35 | 0.32 |
| clip/RN50 | 0.13 | 0.12 | 0.17 | 0.24 | 0.24 | **0.31** | **0.30** | 0.11 | 0.20 | 0.13 | 0.17 | 0.19 | 0.19 | 0.37 | 0.33 | **0.30** | **0.31** | **0.36** | **0.33** |
| v-jepa/vjepa2-vit-giant | 0.16 | 0.16 | 0.20 | 0.24 | **0.25** | 0.28 | 0.24 | 0.16 | **0.21** | 0.16 | **0.20** | **0.23** | 0.16 | **0.41** | **0.40** | 0.29 | 0.28 | 0.31 | 0.30 |
| v-jepa/vjepa2-vit-large | **0.17** | **0.18** | **0.21** | **0.24** | 0.24 | 0.26 | 0.22 | **0.16** | 0.20 | 0.15 | 0.20 | 0.23 | 0.15 | 0.38 | 0.37 | 0.27 | 0.28 | 0.30 | 0.28 |
| v-jepa/vjepa2-vit-giant-384 | 0.16 | 0.18 | 0.20 | 0.23 | 0.23 | 0.26 | 0.23 | 0.15 | 0.19 | **0.16** | 0.19 | 0.20 | 0.14 | 0.40 | 0.38 | 0.27 | 0.26 | 0.29 | 0.28 |

Table S3.3: Model-brain alignment under Linear Predictivity.

| roi / model | V1 | V2 | V3 | V4 | V8 | PIT | FFC | V3A | V3B | V6 | V6A | V7 | IPS1 | MT | MST | FST | LO1 | LO2 | LO3 |
|---|---|---|---|---|---|---|---|---|---|---|---|---|---|---|---|---|---|---|---|
| egomotion | 0.08 | 0.05 | 0.01 | -0.00 | 0.09 | 0.09 | 0.09 | -0.17 | -0.04 | -0.28 | 0.06 | -0.01 | 0.04 | 0.08 | 0.07 | 0.07 | 0.01 | 0.07 | 0.07 |
| room-layout | 0.22 | 0.16 | 0.10 | 0.08 | 0.15 | 0.16 | 0.17 | -0.08 | 0.05 | -0.19 | 0.09 | 0.09 | 0.14 | 0.08 | 0.08 | 0.09 | 0.17 | 0.15 | 0.14 |
| colorization | 0.28 | 0.27 | 0.24 | 0.14 | 0.23 | 0.24 | 0.13 | -0.07 | -0.01 | -0.29 | 0.06 | -0.01 | 0.05 | 0.10 | 0.09 | 0.12 | 0.16 | 0.22 | 0.12 |
| curvature | 0.29 | 0.19 | 0.14 | 0.08 | 0.16 | 0.14 | 0.11 | -0.07 | 0.01 | -0.26 | 0.09 | 0.01 | 0.07 | 0.09 | 0.09 | 0.08 | 0.10 | 0.14 | 0.10 |
| point-matching | -0.01 | 0.11 | 0.07 | 0.05 | 0.09 | 0.07 | 0.05 | -0.15 | 0.02 | -0.27 | 0.04 | 0.02 | 0.08 | 0.09 | 0.08 | 0.07 | 0.03 | 0.08 | 0.08 |
| keypoints2d | 0.08 | 0.05 | 0.04 | -0.02 | -0.01 | 0.05 | 0.00 | -0.11 | -0.07 | -0.31 | -0.02 | -0.05 | 0.00 | 0.04 | 0.04 | 0.02 | 0.04 | 0.04 | 0.03 |
| fixated-pose | 0.07 | 0.02 | -0.02 | -0.00 | 0.08 | 0.10 | 0.10 | -0.16 | -0.01 | -0.37 | 0.06 | 0.04 | 0.11 | 0.05 | 0.05 | 0.04 | 0.04 | 0.09 | 0.10 |
| edge-texture | 0.38 | 0.23 | 0.21 | 0.13 | 0.15 | 0.12 | 0.08 | -0.15 | -0.02 | -0.30 | 0.04 | -0.03 | 0.06 | 0.06 | 0.06 | 0.06 | 0.11 | 0.12 | 0.08 |
| vanishing-point | 0.14 | 0.13 | 0.10 | 0.05 | 0.12 | 0.15 | 0.11 | -0.05 | -0.04 | -0.27 | -0.00 | -0.04 | -0.03 | 0.11 | 0.10 | 0.12 | 0.15 | 0.14 | 0.12 |
| nonfixated-pose | 0.01 | -0.01 | -0.06 | -0.05 | -0.01 | 0.03 | 0.05 | -0.12 | -0.03 | -0.29 | 0.05 | -0.01 | 0.02 | 0.06 | 0.04 | 0.04 | -0.05 | 0.04 | 0.04 |
| normal | 0.26 | 0.24 | 0.27 | 0.22 | 0.29 | 0.32 | 0.23 | 0.04 | 0.11 | -0.22 | 0.16 | 0.09 | 0.16 | 0.20 | 0.16 | 0.16 | 0.29 | 0.28 | 0.24 |
| segment-semantic | 0.34 | 0.26 | 0.31 | 0.25 | 0.32 | 0.31 | 0.24 | 0.07 | 0.15 | -0.09 | 0.18 | 0.11 | 0.22 | 0.18 | 0.17 | 0.16 | 0.31 | 0.31 | 0.23 |
| edge-occlusion | 0.36 | 0.30 | 0.29 | 0.23 | 0.32 | 0.30 | 0.24 | 0.02 | 0.12 | -0.05 | 0.15 | 0.04 | 0.15 | 0.22 | 0.18 | 0.13 | 0.29 | 0.30 | 0.22 |
| inpainting | 0.04 | 0.04 | -0.02 | -0.02 | 0.03 | 0.13 | 0.10 | -0.12 | -0.05 | -0.29 | 0.01 | -0.05 | 0.02 | 0.06 | 0.07 | 0.05 | 0.04 | 0.12 | 0.07 |
| depth-zbuffer | 0.33 | 0.23 | 0.24 | 0.21 | 0.31 | 0.34 | 0.26 | 0.04 | 0.14 | -0.13 | 0.16 | 0.06 | 0.17 | 0.18 | 0.14 | 0.12 | 0.26 | 0.31 | 0.21 |
| denoising | -0.04 | 0.04 | -0.02 | -0.07 | -0.04 | 0.06 | 0.02 | -0.12 | -0.06 | -0.29 | -0.00 | -0.03 | 0.01 | 0.02 | 0.04 | 0.04 | -0.01 | 0.05 | 0.02 |
| autoencoding | -0.07 | -0.02 | -0.05 | -0.07 | -0.05 | 0.03 | 0.01 | -0.12 | -0.07 | -0.28 | -0.00 | -0.04 | 0.00 | 0.01 | 0.03 | 0.02 | -0.03 | 0.03 | 0.01 |
| segment-unsup25d | 0.29 | 0.24 | 0.26 | 0.22 | 0.32 | 0.32 | 0.21 | 0.08 | 0.17 | -0.18 | 0.14 | 0.06 | 0.16 | 0.14 | 0.13 | 0.13 | 0.29 | 0.30 | 0.22 |
| reshading | 0.36 | 0.25 | 0.32 | 0.27 | 0.38 | 0.38 | 0.30 | 0.03 | 0.20 | -0.04 | 0.18 | 0.12 | 0.22 | 0.22 | 0.18 | 0.18 | 0.37 | 0.40 | 0.28 |
| jigsaw | 0.25 | 0.12 | 0.10 | 0.07 | 0.13 | 0.15 | 0.13 | -0.13 | 0.03 | -0.15 | 0.09 | 0.00 | 0.15 | 0.11 | 0.10 | 0.08 | 0.07 | 0.14 | 0.09 |
| depth-euclidean | 0.26 | 0.29 | 0.27 | 0.23 | 0.39 | 0.34 | 0.27 | 0.09 | 0.15 | -0.12 | 0.15 | 0.08 | 0.19 | 0.19 | 0.17 | 0.15 | 0.29 | 0.30 | 0.26 |
| segment-unsup2d | 0.41 | 0.27 | 0.23 | 0.13 | 0.22 | 0.21 | 0.14 | -0.01 | -0.18 | 0.06 | -0.03 | 0.09 | 0.10 | 0.08 | 0.07 | 0.14 | 0.18 | 0.08 | 0.08 |
| keypoints3d | 0.41 | 0.29 | 0.29 | 0.28 | 0.37 | 0.39 | 0.27 | 0.06 | 0.16 | -0.12 | 0.16 | 0.09 | 0.18 | 0.22 | 0.19 | 0.19 | 0.31 | 0.36 | 0.26 |
| class-scene | 0.41 | 0.34 | 0.38 | 0.35 | 0.54 | 0.40 | 0.33 | 0.16 | 0.32 | -0.09 | 0.28 | 0.20 | 0.33 | 0.25 | 0.20 | 0.20 | 0.44 | 0.35 | 0.33 |
| class-object | 0.56 | 0.50 | 0.44 | 0.40 | 0.61 | 0.46 | 0.41 | 0.26 | 0.35 | 0.05 | 0.33 | 0.21 | 0.39 | 0.28 | 0.22 | 0.25 | 0.48 | 0.42 | 0.40 |
| timm/convnext-large | 0.79 | 0.66 | 0.69 | 0.71 | 0.94 | 0.82 | 0.71 | 0.57 | 0.77 | 0.47 | 0.61 | 0.45 | 0.67 | 0.56 | 0.44 | 0.54 | 0.96 | 0.84 | 0.73 |
| clip/ViT-L/14@336px | 0.94 | 0.71 | 0.70 | 0.71 | 0.93 | 0.83 | 0.74 | 0.52 | 0.75 | 0.47 | 0.65 | 0.49 | 0.76 | 0.61 | 0.51 | 0.60 | 0.97 | 0.83 | 0.77 |
| timm/convnext-base | 0.88 | 0.69 | 0.73 | 0.75 | 1.03 | 0.87 | 0.76 | 0.64 | 0.82 | 0.60 | 0.56 | 0.48 | 0.76 | 0.53 | 0.47 | 0.56 | 0.99 | 0.87 | 0.67 |
| clip/ViT-L/14 | 0.87 | 0.66 | 0.70 | 0.69 | 0.95 | 0.81 | 0.76 | 0.45 | 0.80 | 0.60 | 0.61 | 0.43 | 0.75 | 0.60 | 0.53 | 0.63 | 0.91 | 0.84 | 0.75 |
| timm/convnext-small | 0.73 | 0.70 | 0.75 | 0.74 | 0.90 | 0.86 | 0.69 | 0.53 | 0.72 | 0.56 | 0.57 | 0.50 | 0.67 | 0.61 | 0.55 | 0.59 | 1.01 | 0.88 | 0.71 |
| clip/ViT-B/16 | 0.84 | 0.62 | 0.73 | 0.68 | 0.94 | 0.85 | 0.73 | 0.52 | 0.81 | 0.58 | 0.64 | 0.57 | 0.77 | 0.58 | 0.50 | 0.61 | 0.97 | 0.89 | 0.72 |
| v-jepa/vjepa2-vit-huge | 0.74 | 0.79 | 0.87 | 0.82 | 1.08 | 0.90 | 0.68 | 0.98 | 0.98 | 1.04 | 0.92 | 0.76 | 0.71 | 0.77 | 0.77 | 0.68 | 1.10 | 0.89 | 0.80 |
| opt-cwm/opt-cwm | 0.95 | 0.80 | 0.79 | 0.71 | 0.86 | 0.73 | 0.55 | 1.03 | 0.85 | 1.10 | 0.79 | 0.63 | 0.57 | 0.55 | 0.52 | 0.50 | 0.79 | 0.70 | 0.58 |
| mmaction2/swin-tiny | 0.59 | 0.56 | 0.60 | 0.54 | 0.80 | 0.68 | 0.55 | 0.51 | 0.56 | 0.35 | 0.47 | 0.34 | 0.52 | 0.56 | 0.46 | 0.55 | 0.74 | 0.69 | 0.62 |
| timm/convnext-tiny | 0.78 | 0.70 | 0.72 | 0.66 | 0.84 | 0.77 | 0.69 | 0.47 | 0.65 | 0.21 | 0.53 | 0.39 | 0.67 | 0.59 | 0.49 | 0.55 | 0.91 | 0.82 | 0.73 |
| mmaction2/mvit-small | 0.92 | 0.76 | 0.75 | 0.62 | 0.84 | 0.75 | 0.68 | 0.60 | 0.63 | 0.48 | 0.55 | 0.36 | 0.64 | 0.60 | 0.54 | 0.56 | 0.66 | 0.72 | 0.66 |
| mmaction2/timesformer | 0.32 | 0.29 | 0.27 | 0.27 | 0.45 | 0.37 | 0.33 | 0.19 | 0.28 | 0.07 | 0.27 | 0.18 | 0.31 | 0.24 | 0.19 | 0.25 | 0.40 | 0.36 | 0.33 |
| timm/resnet18 | 0.81 | 0.62 | 0.68 | 0.69 | 0.97 | 0.93 | 0.80 | 0.62 | 0.80 | 0.47 | 0.66 | 0.58 | 0.78 | 0.64 | 0.54 | 0.66 | 1.03 | 0.86 | 0.77 |
| vissl/simclr-resnet101 | 0.99 | 0.72 | 0.79 | 0.78 | 1.07 | 0.95 | 0.85 | 0.79 | 0.86 | 0.72 | 0.72 | 0.62 | 0.85 | 0.65 | 0.57 | 0.65 | 1.08 | 0.95 | 0.82 |
| clip/ViT-B/32 | 0.95 | 0.76 | 0.80 | 0.81 | 1.16 | 1.04 | 0.94 | 0.80 | 0.93 | 0.83 | 0.82 | 0.64 | 0.90 | 0.73 | 0.61 | 0.72 | 1.21 | 1.04 | 0.92 |
| vissl/simclr-resnet50 | 0.97 | 0.70 | 0.77 | 0.78 | 1.07 | 0.97 | 0.85 | 0.77 | 0.85 | 0.72 | 0.71 | 0.55 | 0.81 | 0.65 | 0.55 | 0.62 | 1.04 | 0.96 | 0.82 |
| timm/resnet50 | 0.78 | 0.59 | 0.64 | 0.71 | 1.02 | 0.98 | 0.85 | 0.55 | 0.86 | 0.63 | 0.70 | 0.59 | 0.87 | 0.71 | 0.61 | 0.69 | 1.12 | 0.97 | 0.86 |
| vggt/VGGT-1B | 0.95 | 0.75 | 0.81 | 0.84 | 1.14 | 1.09 | 1.01 | 0.92 | 1.10 | 1.01 | 0.96 | 0.75 | 1.03 | 0.75 | 0.67 | 0.75 | 1.27 | 1.09 | 0.99 |
| clip/RN50 | 0.80 | 0.72 | 0.83 | 0.89 | 1.20 | **1.11** | **1.02** | 0.84 | 1.12 | 0.88 | 0.89 | 0.70 | **1.04** | 0.81 | 0.71 | **0.79** | **1.33** | **1.15** | **1.01** |
| v-jepa/vjepa2-vit-giant | 0.95 | 0.88 | 0.98 | 0.91 | **1.23** | 1.01 | 0.81 | 1.21 | **1.12** | 1.06 | **1.04** | **0.88** | 0.83 | **0.88** | **0.88** | 0.77 | 1.19 | 0.96 | 0.93 |
| v-jepa/vjepa2-vit-large | **1.10** | **0.97** | **1.00** | **0.92** | 1.18 | 0.95 | 0.74 | **1.26** | 1.08 | **1.21** | 0.94 | 0.86 | 0.79 | 0.81 | 0.81 | 0.72 | 1.18 | 0.94 | 0.87 |
| v-jepa/vjepa2-vit-giant-384 | 0.93 | 0.96 | 0.95 | 0.87 | 1.17 | 0.94 | 0.77 | 1.18 | 1.06 | 1.13 | 0.95 | 0.77 | 0.74 | 0.85 | 0.84 | 0.73 | 1.11 | 0.91 | 0.87 |

Table S3.4: Model-brain alignment under Linear Predictivity, normalized to NeuroAI Turing Test.

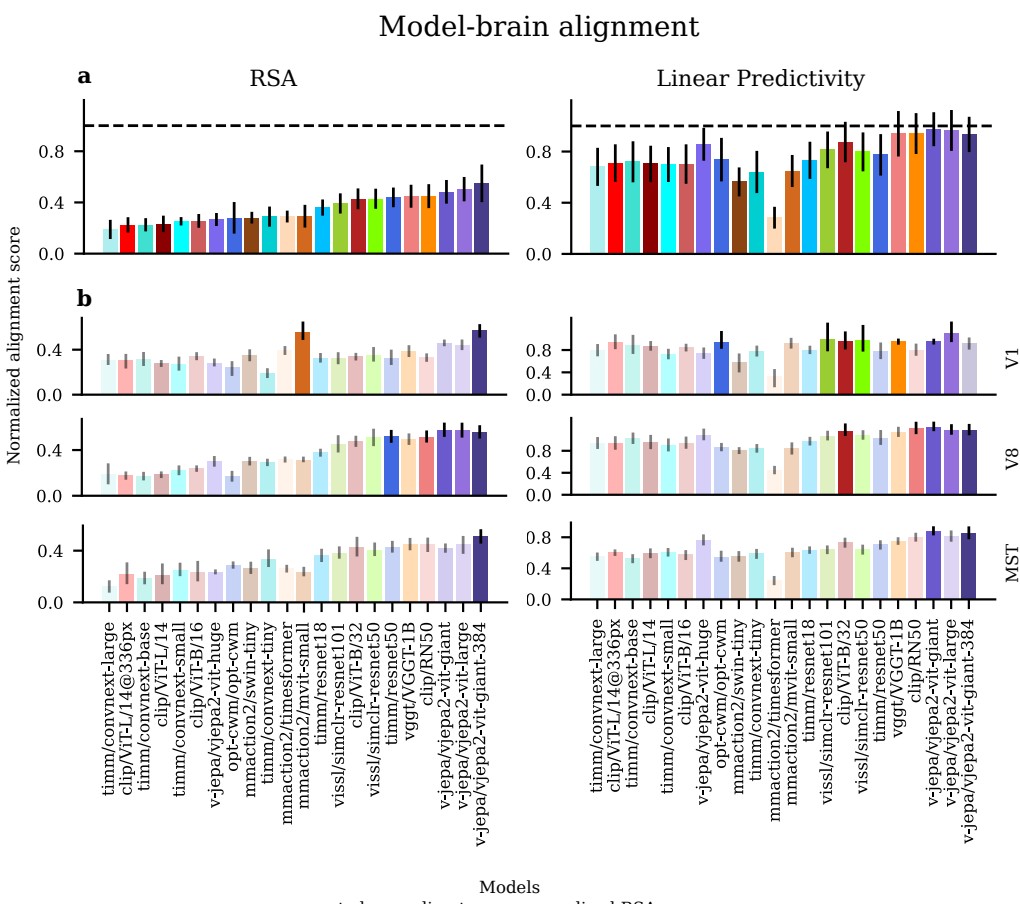

Figure S3.2: Normalizing alignment scores shows that the benchmark is saturated for LP but not for RSA. Legend see main Figure 2

Alignment patterns

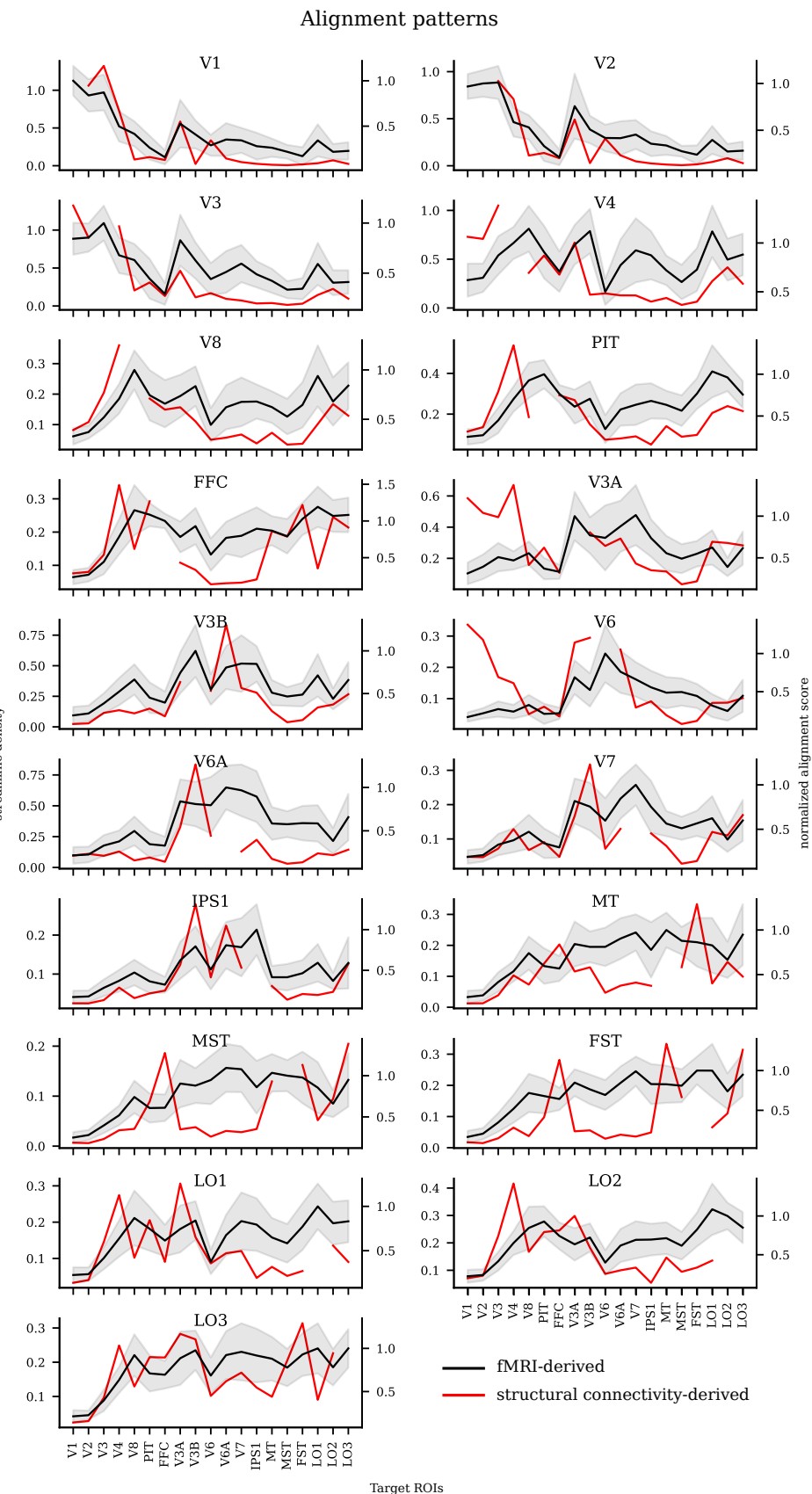

Figure S3.3: RSA-based fMRI-derived and structural connectivity-derived alignment patterns.

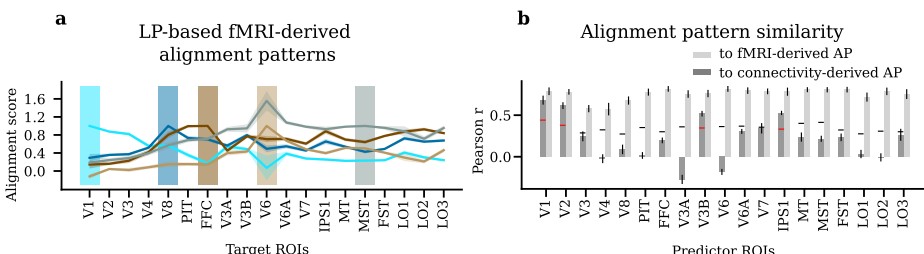

Figure S3.4: Brain-brain alignment patterns analysis for LP-based fMRI-derived and structural connectivity-derived alignment patterns. Legend see main Figure 3

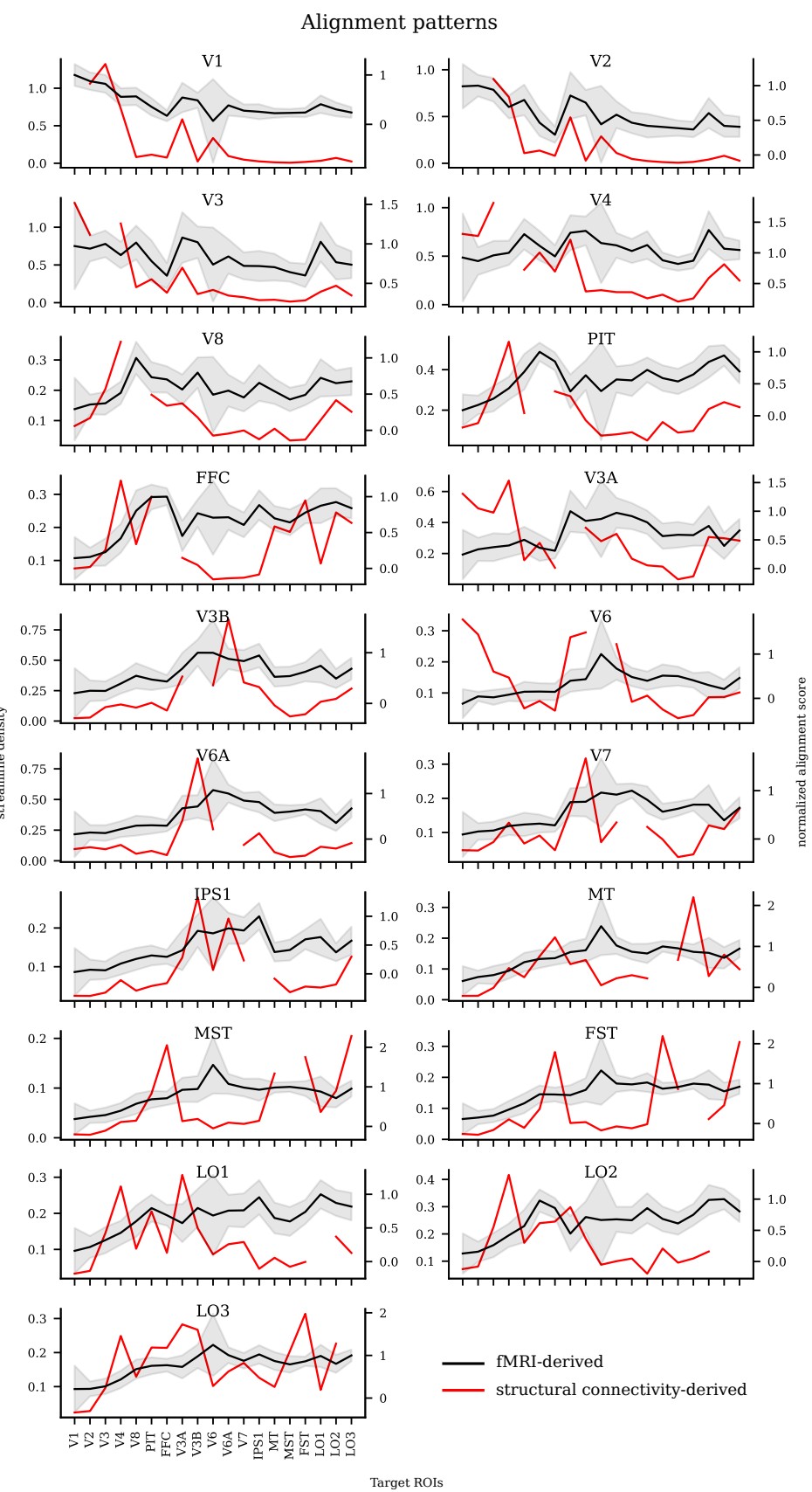

Figure S3.5: LP-based fMRI-derived and structural connectivity-derived alignment patterns.

Figure S3.6: Model-brain alignment pattern analysis for LP-based alignment patterns. Legend see main Figure 4

| | fMRI-derived APS | Connectivity-derived APS | Random connectivity (95th percentile) | Random connectivity (5th percentile) |
|---|---|---|---|---|
| V1 | 0.91 | 0.82 | 0.50 | -0.28 |
| V2 | 0.90 | 0.80 | 0.47 | -0.27 |
| V3 | 0.81 | 0.59 | 0.33 | -0.28 |
| V4 | 0.68 | 0.06 | 0.29 | -0.29 |
| V8 | 0.68 | 0.15 | 0.26 | -0.30 |
| PIT | 0.79 | 0.20 | 0.30 | -0.36 |
| FFC | 0.80 | 0.34 | 0.29 | -0.38 |
| V3A | 0.76 | -0.16 | 0.35 | -0.28 |
| V3B | 0.81 | 0.55 | 0.33 | -0.33 |
| V6 | 0.83 | 0.04 | 0.36 | -0.30 |
| V6A | 0.82 | 0.39 | 0.34 | -0.32 |
| V7 | 0.82 | 0.47 | 0.33 | -0.32 |
| IPS1 | 0.82 | 0.66 | 0.35 | -0.32 |
| MT | 0.76 | 0.27 | 0.26 | -0.35 |
| MST | 0.78 | 0.24 | 0.29 | -0.36 |
| FST | 0.82 | 0.27 | 0.28 | -0.39 |
| LO1 | 0.72 | 0.25 | 0.23 | -0.36 |
| LO2 | 0.80 | 0.22 | 0.31 | -0.37 |
| LO3 | 0.74 | 0.40 | 0.24 | -0.38 |

Table S3.5: Brain-brain alignment pattern similarity (RSA-based) as shown in Fig. 3.

| | fMRI-derived APS | Connectivity-derived APS | Random connectivity (95th percentile) | Random connectivity (5th percentile) |
|---|---|---|---|---|
| V1 | 0.79 | 0.68 | 0.40 | -0.24 |
| V2 | 0.78 | 0.61 | 0.34 | -0.25 |
| V3 | 0.58 | 0.25 | 0.26 | -0.23 |
| V4 | 0.57 | -0.02 | 0.28 | -0.24 |
| V8 | 0.68 | 0.09 | 0.25 | -0.26 |
| PIT | 0.78 | 0.02 | 0.30 | -0.32 |
| FFC | 0.82 | 0.20 | 0.28 | -0.35 |
| V3A | 0.75 | -0.27 | 0.32 | -0.29 |
| V3B | 0.76 | 0.52 | 0.30 | -0.31 |
| V6 | 0.82 | -0.18 | 0.31 | -0.32 |
| V6A | 0.80 | 0.30 | 0.32 | -0.30 |
| V7 | 0.79 | 0.34 | 0.30 | -0.31 |
| IPS1 | 0.78 | 0.53 | 0.29 | -0.32 |
| MT | 0.81 | 0.24 | 0.33 | -0.34 |
| MST | 0.81 | 0.22 | 0.33 | -0.34 |
| FST | 0.81 | 0.23 | 0.29 | -0.36 |
| LO1 | 0.72 | 0.03 | 0.24 | -0.34 |
| LO2 | 0.79 | -0.01 | 0.28 | -0.36 |
| LO3 | 0.75 | 0.26 | 0.26 | -0.35 |

Table S3.6: Brain-brain alignment pattern similarity (LP-based) as shown in Fig S3.4.

### S3.0.1 ALIGNMENT PATTERN SIMILARITY FOR TRAINED VS. RANDOMLY INITIALISED MODELS

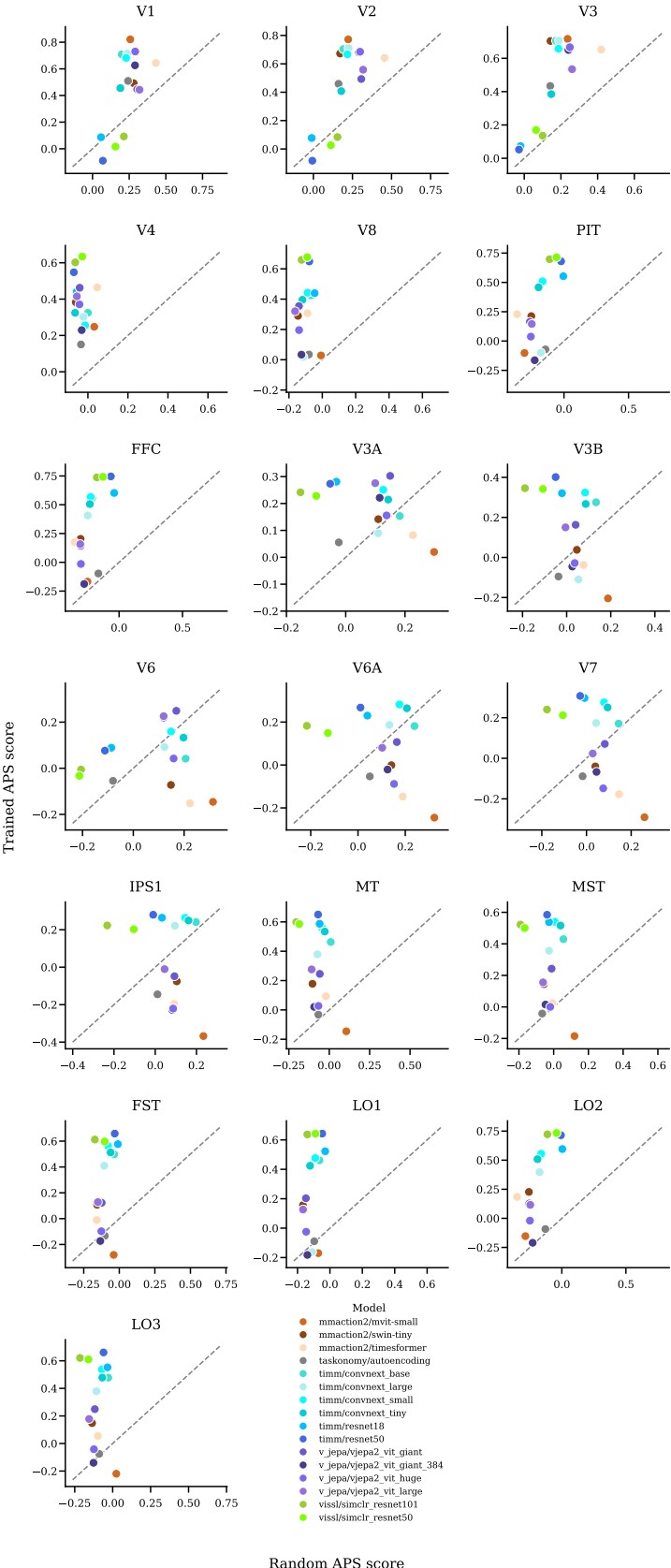

Figure S3.7: RSA-based APS for trained vs. untrained models.

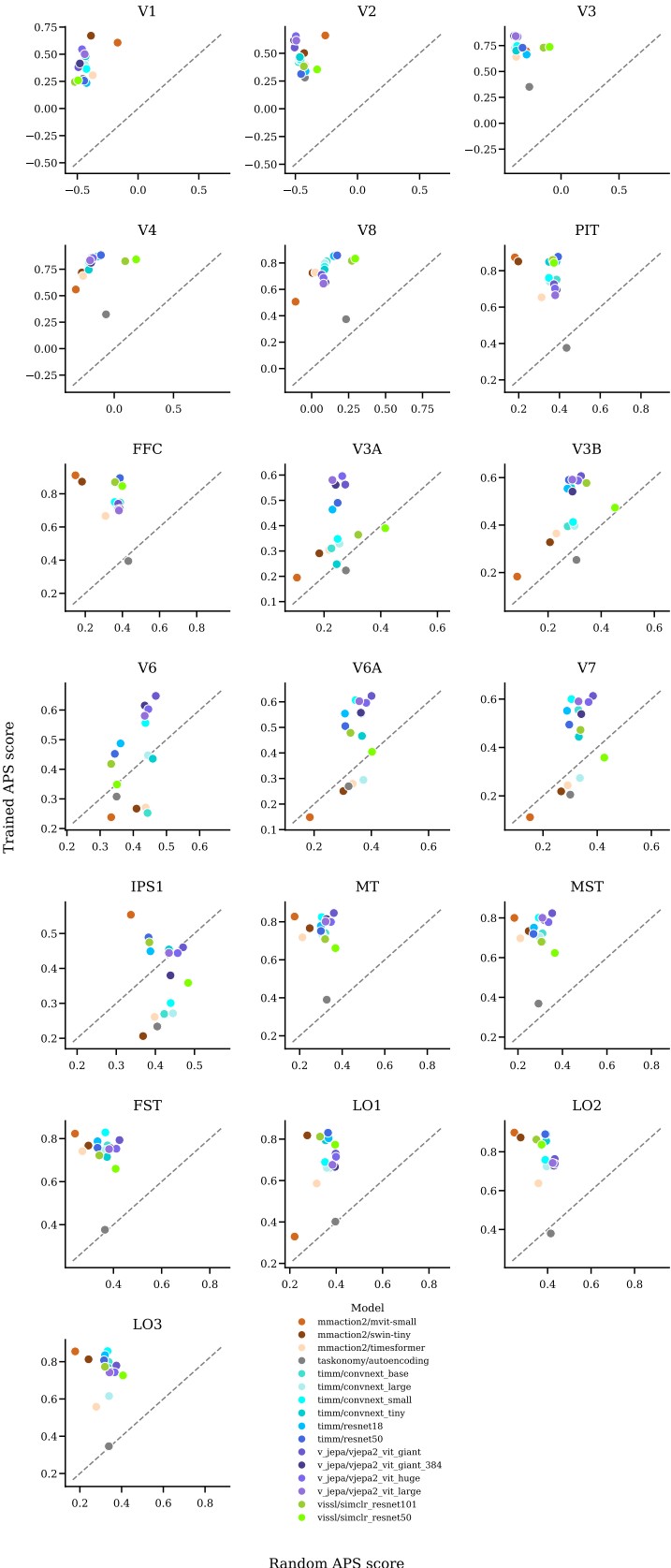

Figure S3.8: LP-based APS for trained vs. untrained models.

## S3.1 ALIGNMENT SCORES AND ALIGNMENT PATTERN SIMILARITY ACROSS ALL LAYERS

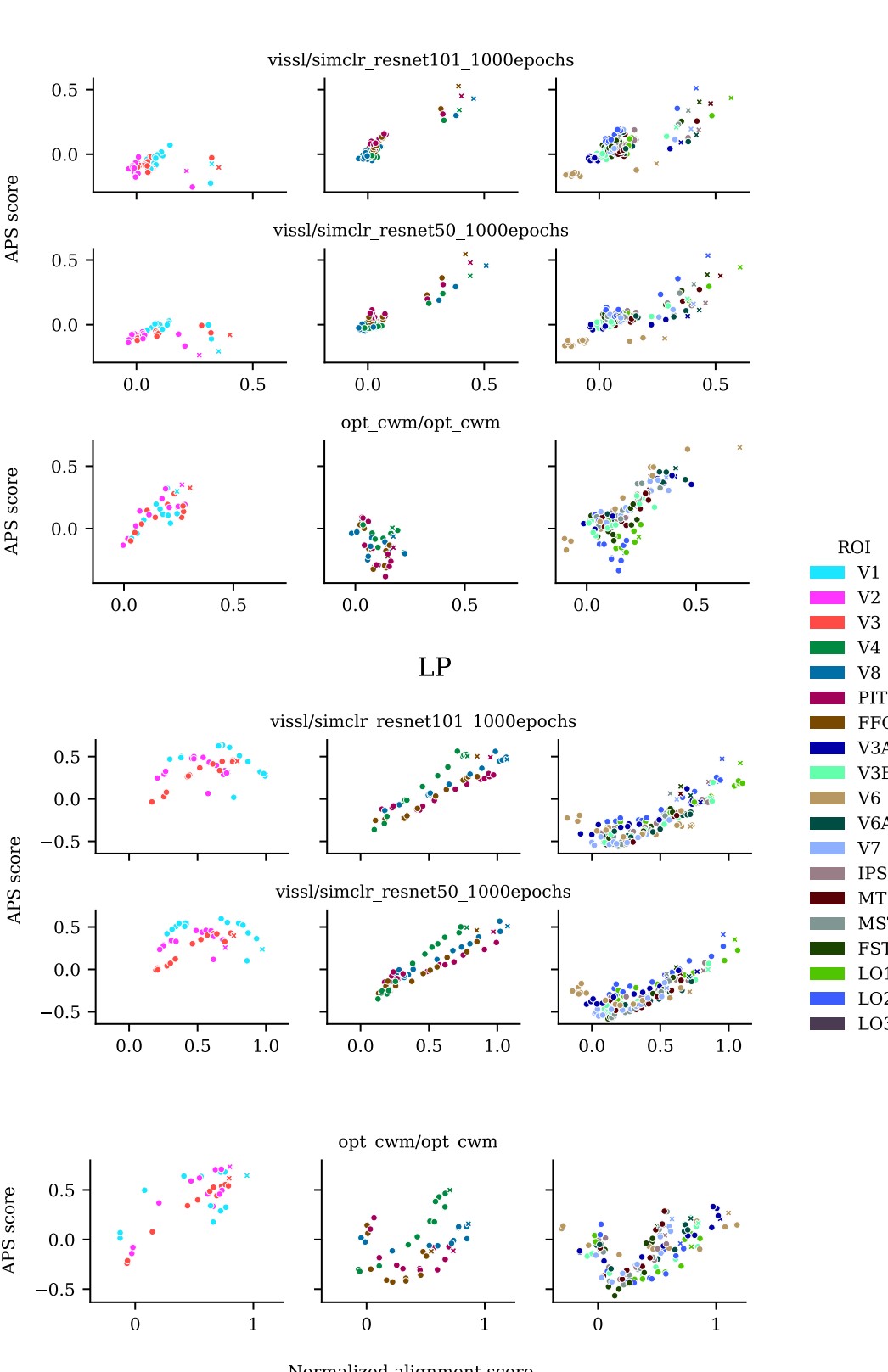

Figure S3.9: Scores (alignment and APS) for all layers of all models from the VISSL model family as well as the OPT-CWM and all ROIs. Top panels show RSA, bottom panels show LP. Stars indicate best layers selected based on alignment score.

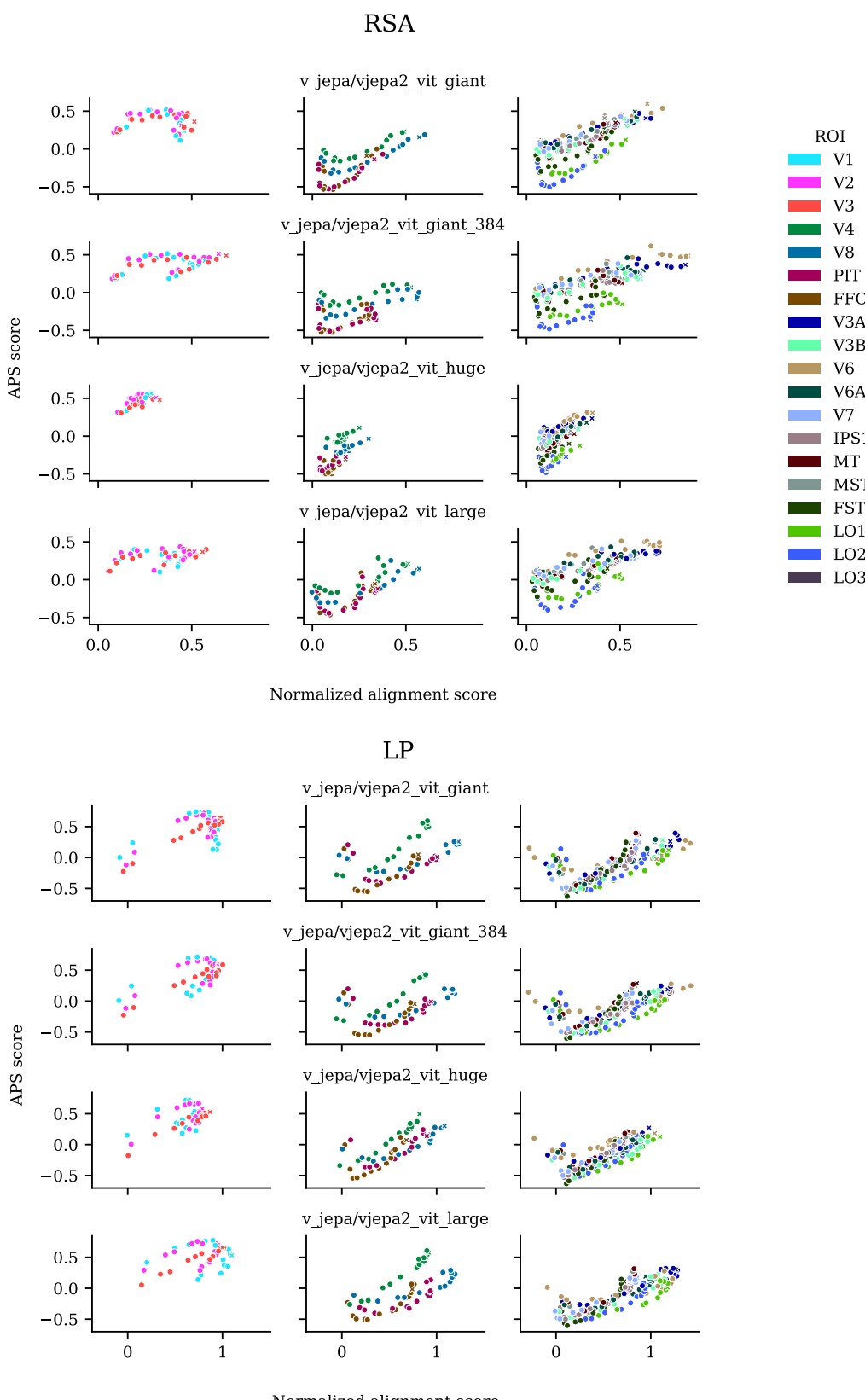

Figure S3.10: Scores (alignment and APS) for all layers of all models from the V-JEPA model family and all ROIs. Legend see Fig. S3.9.

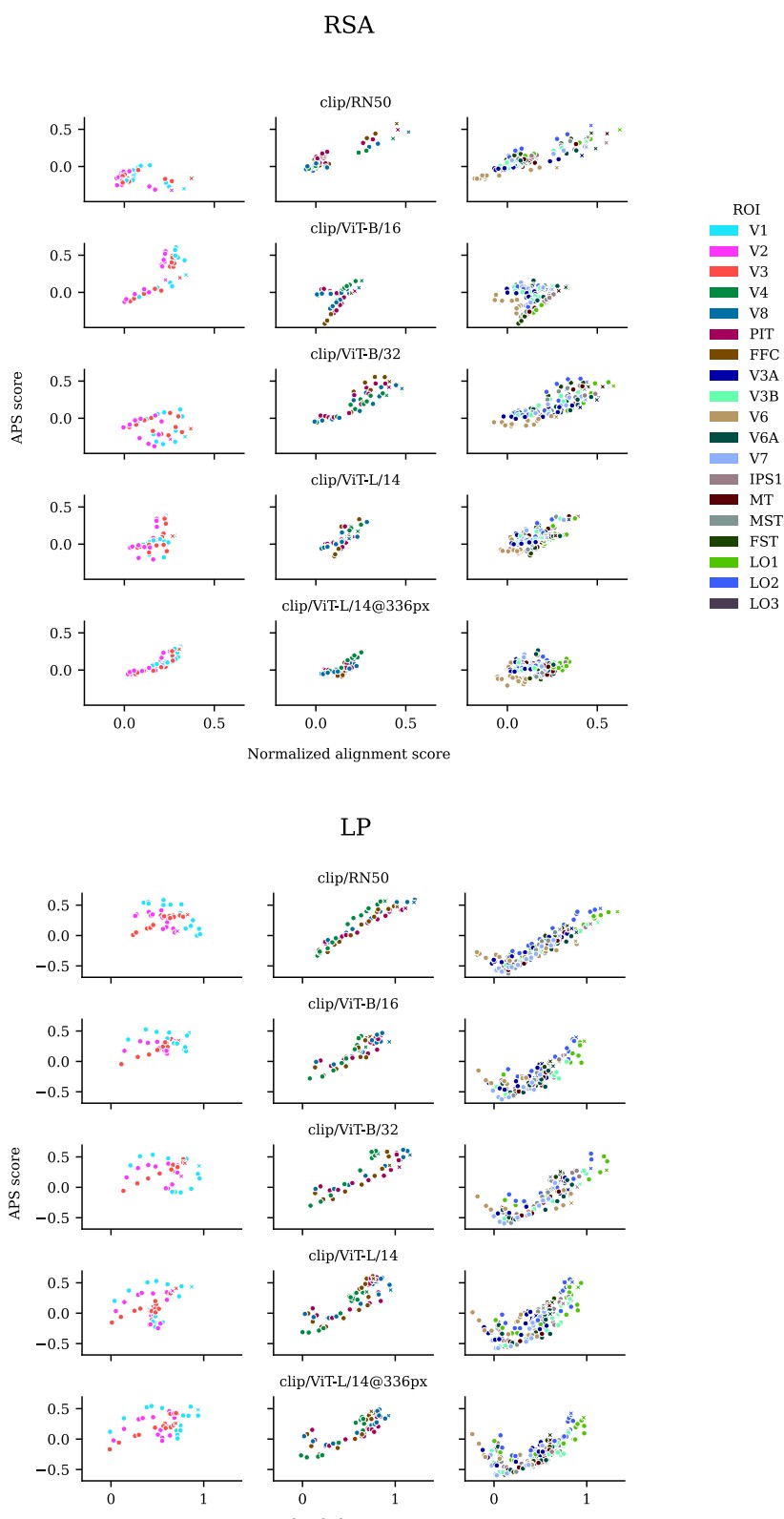

Figure S3.11: Scores (alignment and APS) for all layers of all models from the CLIP model family and all ROIs. Legend see Fig. S3.9.

