# OpenReview forum: "Only Brains Align with Brains: Cross-Region Alignment Patterns Expose Limits of Normative Models"
_ICLR.cc/2026/Conference — ICLR 2026 Poster_

### Official Review · Reviewer_yHdD · 2025-10-31

**Soundness:** 3
**Presentation:** 2
**Contribution:** 3
**Rating:** 6
**Confidence:** 3

**Summary:**

The paper benchmarks 47 image and video models on BOLD-Moments fMRI with RSA and linear predictivity, finding that rankings depend on the metric and many models are practically equivalent within subject variability. It proposes Alignment Pattern Similarity, grounded in structural connectivity, to test whether models preserve each ROI’s cross-region similarity profile, and reports that brains align with brains while models generally fail to match these patterns, arguing for stricter anatomy-informed evaluation.

**Strengths:**

- The authors raise a very important and timely issue regarding AI-model alignment.
- The authors propose a solution that I think is quite novel, which is checking the inter-regional similarities between brain regions and with one region replaced by its best-fit model.
-  Showing that most models fall within subject-level variability is a much-needed report.
-  I personally find the high APS for brain-brain alignment very interesting (but could the author please provide the baseline elaborated in the weaknesses section below?).

**Weaknesses:**

- I have a philosophical, high-level concern. I am not sure if requiring the model representation (that has a best fit to one brain region) to have relationships to other brain regions in a way that one brain region relates to those brain regions is an unnecessarily strict requirement.  But I do think it is still an interesting thing to check and report, which could be the author’s point.

- In this line of thought, I think another interesting analysis would be
1. First, (within a single model) measure the similarity between layers that have high alignments to brain regions of interest. Call that s’_model_RSA(i,j) (or s’_model_LP) for brain regions i and j. For example, say layer 3 corresponds to ROI 1 of the brain, and layer 5 corresponds to ROI 2 of the brain. Then s’_model_RSA(1,2) measures the similarity between layer 3 and layer 5 of the networks.
2. Measure and report R^2 between S’_model_RSA and brain-to-brain S_RSA (the latter one is from the paper e.g. Figure 5d top left red line).
Would it be easy to perform this analysis? This is less strict than the author’s requirement. Qualitatively speaking, it is like saying we don’t care exactly how layers talk to each other (and how the ROIs talk to each other), and the only thing we care about is the relationship (RSA/LP value) between these layers and the relationship (RSA/LP value) between these ROIs. We might want these relationships to be similar (high R^2) between the model and the brain.

- Perhaps more importantly, I think the authors should report the baseline for APS, where the ROI connectivity graph is randomly generated.

- I understand that the paper's idea and methodology are not easy to explain, but I think the clarity of the paper can be dramatically improved. Figure 1 is well-intended, but it only makes sense after reading the entire paper. As a reader, it is especially unclear what the cartoon plots under "Alignment Patterns" indicate (one would wonder that the x-axis and y-axis are), unless they have read section 3.5 (which I don't think is also clearly written). I am not clear what the best way is to explain the overall pipeline, but I don't think Figure 1 helps much at all.  In section 3.5, the last sentence was nearly impossible to understand.

**Questions:**

Please see the weaknesses section for the question.

---

> ### Author Response · Authors · 2025-11-26
> **Response to reviewer**
>
> We thank the reviewer for their time in reviewing our paper, and for highlighting the **importance and timeliness** of the issues we raise regarding model-brain alignment benchmarking practices, as well as the **novelty** of our approach (alignment pattern analysis).
>
> We also thank the reviewer for their concerns and suggestions, which we address below.
>
> **Philosophical concern: APS may be unnecessarily strict**
>
> Thank you for raising this conceptual point. We have refined our storyline to make an explicit distinction between two complementary goals: searching for **brain-predictive** models, and searching for **brain-aligned** models. Existing pipelines are well-suited for identifying the most brain-predictive models, and we agree that for this goal, alignment pattern similarity is not a requirement. However, when the question is whether a model is **brain-aligned** in a stricter sense which allows to draw conclusions about the inductive biases shaping the visual system, we argue that it is reasonable to require that the model replicate brain–brain alignment patterns. This argument is supported by the fact that one human brain is aligned to other human brains under the same criterion (for most visual regions; [brain-brain alignment patterns](https://ibb.co/W4jgwxm5); [alignment pattern similarity distribtuions](https://ibb.co/gZgHrqP9)), which demonstrates that the APS constraint is attainable and not too strict in practice. While we are incorporating the distinction between brain-predictivity and brain-alignment into the revised manuscript, we kindly ask the reviewer to refer to the storyline uploaded as part of the **Supplementary Material**.
>
> **Suggested new analysis**
>
> Thank you for this interesting suggestion. A closely related idea has been explored in prior work - for example, Nonaka et al. (2021)[1] proposed the Brain Hierarchy score to quantify the correspondence between model layer hierarchies and the brain’s visual hierarchy. We will explicitly reference and discuss this line of work in the revised manuscript.
> Performing the analysis would require implementing and running a new variant of the benchmark, where we would evaluate alignment between **each pair of layers within a model** (instead of evaluating alignment between model layers and brain ROIs). We are concerned that this additional analysis may complicate the narrative and obscure the main conceptual contribution. That said, if the reviewer believes that including this analysis would meaningfully strengthen the paper, we would be happy to include the results in the appendix.
>
> **Baseline with random connectivity**
>
> Concerning the comparison to connectivity-derived alignment patterns, we are currently re-running the analysis, replacing our minimal connectivity graph with structural connectivity matrices from the Human Connectome Project. We will provide results including a random connectivity baseline within the rebuttal period.
>
> **Clarity of the paper**
>
> We thank the reviewer for this comment. We agree that the clarity of our paper can be improved. We have prepared a revised [overview figure](https://ibb.co/3ymF9M0J) that illustrates our approach and positions it relative to the NeuroAI Turing Test [2], and we are revising our paper to reflect the storyline that we have, for now, uploaded as part of the **Supplementary Material**. This also includes a revised version of the Methods section that details the alignment pattern analysis (previously section 3.5), with a clear and consistent notation. We would appreciate the reviewer's feedback on these improvements.
>
> [1]: Nonaka, Soma, et al. "Brain hierarchy score: Which deep neural networks are hierarchically brain-like?." IScience 24.9 (2021).
>
> [2]: Feather, Jenelle, et al. "Brain-model evaluations need the neuroai turing test." arXiv preprint arXiv:2502.16238 (2025).

---

### Official Review · Reviewer_NsfX · 2025-11-01

**Soundness:** 2
**Presentation:** 2
**Contribution:** 2
**Rating:** 4
**Confidence:** 3

**Summary:**

This paper evaluates 47 vision models on the BOLDMoments video fMRI dataset using RSA and linear predictivity (LP) to assess brain-model alignment. The authors identify V-JEPA2 as achieving strongest alignment but show that many models are "practically equivalent" within subject-level variability. They introducxe Alignment Pattern Similarity (APS), a novel metric that compares cross-region alignment patterns to anatomical connectivity patterns. The key finding: while normative models can align well with individual brain regions, they fail to reproduce brain-to-brain cross-region alignment patterns, revealing a fundamental limitation of current approaches.

**Strengths:**

1. **Important conceptual contribution:** APS is a genuinely novel idea that addresses a real gap in the field. The insight that models should preserve relational structure across regions (not just match individual regions) is valuable and biologically motivated.
2. **Comprehensive model evaluation:** Testing 47 models spanning diverse architectures (CNNs, Transformers), objectives (supervised, self-supervised, multimodal), and modalities (image, video) on a large video dataset is thorough.
3. **Strong empirical results:** The finding that brain-to-brain APS is high while model-to-brain APS is near-zero is an interesting result.

**Weaknesses:**

1. **Figure quality is severely lacking:** The sketch-based design of figure 1 is not ideal for a publication at a top tier conference. I would recommend redoing it with vector graphics. The other figures have almost illegible axis labels with overlapping/compressed model and ROI names.
2. **Limited to RSA and LP Despite Known Limitations:** Paper correctly identifies that "conclusions depend strongly on choice of metric" yet only uses RSA and LP. Missing modern metrics like CKA[1], RTD[2] and NSA[3]
3. **APS Validation is Incomplete:** No null model is presented. What is expected APS for random alignment patterns? No p-values or significance tests. Could spatial proximity or retinotopy explain results instead of connectivity? Does APS generalize to NSD or HCP 7T?
4. **Model-Brain APS Results Are Hard to Interpret:** Low APS could mean: (a) models fundamentally limited, (b) wrong layer selection, or (c) APS too strict but the authors assert (a) without ruling out the other options. Layer-wise analysis of APS is also missing.

**Questions:**

Please refer to weaknesses.

---

> ### Author Response · Authors · 2025-11-26
> **Response to reviewer [1]**
>
> We thank the reviewer for their time in reviewing our paper and for highlighting the **importance and conceptual novelty** of our core contribution (alignment pattern analysis), as well as the **comprehensiveness of our benchmark** and the **strength of our empirical results**.
>
> We also thank the reviewer for raising their concerns, which we address below.
>
> **Figure quality**
>
> Thank you for this helpful comment. We agree that the figure quality can be improved. We are currently redesigning all figures for the revised manuscript. To illustrate the planned improvements, we have already completed a redesigned version of the [overview figure](https://ibb.co/3ymF9M0J). Figures showing new results, e.g. on the [distribution of alignment pattern similarities](https://ibb.co/gZgHrqP9) illustrate the style in which we will update all remaining figures (higher-resolution vector elements, clearer typography, and non-overlapping labels).
>
> **Including additional alignment measures**
>
> Thank you for raising this point. Based on all the reviewers’ feedback, we have clarified the core contributions of the paper and substantially reworked the storyline in the revised version (please refer to the uploaded **Supplementary Material** and the [Official Comment](https://openreview.net/forum?id=cMGJcHHI7d&noteId=VqzgRIxOEB) for the storyline).
> Briefly, the central aim of the work is not to provide a broad comparison of alignment metrics, but rather to introduce a new relational criterion for brain-alignment. This criterion is compatible with, and applicable across, the commonly used alignment metrics in the field. Given this clarified framing, our interpretation is that the main contribution does not depend on adding more metrics beyond RSA and LP. However, we would appreciate the reviewer’s opinion: do you believe that including additional metrics such as CKA would strengthen the paper *without detracting from clarity*? If so, we would be happy to include the key analyses using these metrics in the appendix of the revised manuscript.
>
> **APS validation**
>
> Thank you for raising these important points. We have substantially re-conceptualised the alignment pattern analysis. Crucially, the core of the analysis now rests on comparing fMRI-derived brain-brain alignment patterns with model-feature-derived model-brain alignment patterns directly. This is explained in the Methods section of the revised manuscript, which we already uploaded as part of the **Supplementary Material**. We validate the results ([alignment patterns](https://ibb.co/W4jgwxm5); [distribution of alignment pattern similarities](https://ibb.co/gZgHrqP9)) of this analysis in a number of ways.
>
> *Empirical distribution of brain-brain alignment patterns and their similarities*
>
> We generate an empirical distribution of brain-brain alignment patterns by predicting each participant’s alignment pattern from a single other participant, repeated five times to generate five independent patterns per subject. We then generate a distribution of alignment pattern similarities by calculating the alignment pattern similarity (APS) between each individual alignment pattern for a given subject and the average alignment pattern that results from averaging all independent alignment patterns, i.e. those in which this subject did not participate (this procedure is detailed in Section 2.1 of the Methods section provided as part of the **Supplementary Material**).
> This provides a robust empirical distribution of expected APS values across subjects and clarifies the level of inherent inter-individual variability against which model–brain APS should be interpreted.
>
> *Null model for model–brain APS using networks with random weights; spatial proximity and retinotopy*
>
> For the comparison between fMRI-derived brain-brain alignment patterns and model-feature-derived model-brain alignment patterns, we added a baseline by evaluating the same architectures with random weights as predictors of fMRI responses. This preserves architectural spatial structure (including any implicit retinotopy-like biases) while removing learned feature structure. These random networks yield consistently lower APS values than their trained counterparts, demonstrating that the model-brain APS effects cannot be explained by architectural spatial priors alone. Result figures: [alignment patterns for models with random weights](https://ibb.co/bjVszGz1); [distribution of APS trained vs. untrained](https://ibb.co/jPX3pbCF). The reviewer may note that this analysis is restricted to a subset of model libraries where it was rather straightforward to run models with random weights. If the reviewer thinks it would add to the paper to also do this analysis for the remaining model libraries, we would be happy to include this in the revised version of the paper.
>
> We also would like to ask the reviewer to clarify whether this addresses their concerns about **spatial proximity and retinotopy** as confounding factors.

---

> ### Author Response · Authors · 2025-11-26
> **Response to reviewer [2]**
>
> **APS validation - continued**
>
> *Random baseline for connectivity-derived alignment pattern similarity*
>
> Concerning the comparison to connectivity-derived alignment patterns, we are currently re-running the analysis, replacing our minimal connectivity graph with structural connectivity matrices from the [Human Connectome Project](https://brainlife.io/pub/640a3f9dc538c16a826f9b1a?utm_source=chatgpt.com#release.1.preprocessing). We will provide results including a random connectivity baseline within the rebuttal period.
>
> **Scope of generalization across datasets**
>
> We appreciate the suggestion to evaluate APS generalization across additional fMRI datasets (e.g., NSD, HCP 7T). While this is an important direction for future work, it falls outside the scope of the current manuscript. Our primary goal is to introduce APS and provide a deep characterization of its behavior on a single large, high-temporal-resolution video fMRI dataset. We now clarify this positioning explicitly in the text.
>
> **Interpretation of APS**
>
> Thank you for this thoughtful comment. Based on the reviewer’s feedback, we refined the storyline to more clearly articulate the question our work aims to address. Importantly, we suggest to distinguish two complementary aims: searching for **brain-predictive** models (for which benchmarking pipelines are well-suited) and searching for **brain-aligned** models. In the latter case, we argue that the relational constraint captured by APS is not too strict; rather, it reflects a meaningful property of the biological system that a brain-aligned model should ideally capture. Importantly, we now show that a single subject’s brain satisfies this criterion: one brain is aligned, in this relational sense, to other brains for most visual regions. This demonstrates that APS is not an unrealistically strong requirement but an attainable one, met by the biological system itself.
> While we are incorporating the distinction between **brain-predictivity** and **brain-alignment** into the revised manuscript, we kindly ask the reviewer to refer to the storyline uploaded as part of the **Supplementary Material**.
>
>
> *Concerning layer-selection*: we perform layer selection based on alignment scores, as is common in brain-alignment benchmarking pipelines, and then evaluate alignment pattern similarity for the best-performing layers in the above sense. We interpret the reviewer’s comment to suggest that layers should be selected based on APS, and models’ brain-alignment in the alignment-pattern sense should be recalculated for layers selected in this way. This would amount to redoing the benchmark using APS as the measure for brain-alignment. We would like to ask the reviewer to confirm our interpretation (or otherwise please clarify), and also comment on whether you think this would be in scope of the paper without subtracting from the clarity of the storyline.

---

### Official Review · Reviewer_jG2B · 2025-11-04

**Soundness:** 3
**Presentation:** 2
**Contribution:** 3
**Rating:** 4
**Confidence:** 4

**Summary:**

This paper benchmarks tens of image and video models on the BOLD-Moments fMRI dataset using Representational Similarity Analysis (RSA) and Linear Predictivity (LP), argues that many models are practically equivalent despite their rankings because their model→brain alignment falls within the distribution of brain↔brain alignment, and introduces Alignment Pattern Similarity (APS)—a connectivity-grounded test of whether a model preserves each ROI’s cross-region similarity pattern. The key claim is that, although some models score well on RSA/LP, they generally fail to reproduce brain-to-brain cross-region patterns under APS, whereas brains do.

**Strengths:**

- The paper tackles the growing concern that conclusions about “brain alignment” depend on metric choice, and proposes a relational criterion that goes beyond local fits.
- The practical-equivalence analysis is a strong way to assess when model-ranking differences are statistically meaningful.
- Creative attempt to incorporate anatomical priors and cross-region dependencies into model evaluations.
- It's nice that the paper includes a diverse set of image and video models across supervised and self-supervised objectives.

**Weaknesses:**

- My biggest concern is that the three aims—(1) benchmarking image/video models on the BOLD Moments dataset, (2) demonstrating practical equivalence of several models (in the sense that they lie within the brain-brain similarity range), and (3) introducing APS—feel only loosely connected. Each of these components could stand as an independent contribution, but when presented together, the narrative lacks a clear causal or conceptual throughline. For instance, the benchmark and equivalence analyses establish the limitations of RSA/LP metrics but are not explicitly framed as motivating APS; instead, APS appears as a parallel idea rather than a methodological extension.
- The practical-equivalence conclusion hinges on the brain↔brain baseline, which is constructed by averaging voxel responses across subjects before comparing to a held-out subject. This raises concerns about functional alignment (are voxels/topographies aligned across individuals?) and the arbitrariness of cross-subject averaging. The authors should: (i) report within-subject baselines, (ii) test hyperalignment or response-based alignment, and (iii) evaluate subject→subject prediction without averaging. Related ideas have appeared in the NeuroAI Turing Test (Feather et al., 2025), which formalizes a similar distributional criterion for model evaluation; citing and differentiating from that work would clarify novelty.
- The phrase “shifting the focus from single-metric rankings to the stability of rankings” is confusing, since two metrics (RSA, LP) are still used. The authors actually examine whether model orderings are stable given subject variability—i.e., when multiple models are within the brain↔brain alignment range. This should be explicitly defined early to avoid misinterpretation.
- The statement that “recent benchmarks of video models have relied on a single alignment metric, while metric-comparison studies rarely include modern video architectures” should be supported with citations

**Questions:**

- How sensitive are the practical-equivalence findings to the brain↔brain baseline (averaged vs. individual subject predictors; with/without hyperalignment; within-subject splits)?
- Could you compare your equivalence criterion to the NeuroAI Turing Test (Feather et al., 2025) to delineate conceptual differences and overlapping assumptions?

---

> ### Author Response · Authors · 2025-11-26
> **Response to Reviewer**
>
> We thank the reviewer for their time in reviewing our paper and for highlighting the **relevance** of *tackling growing concerns about brain-alignment methodology*, as well as the **novelty and soundness** of our approach of *applying a relational criterion to assess brain-alignment of a large range of state-of-the-art vision models*.
>
> We also thank the reviewer for raising their concerns, which we address below
>
> **Missing connection between findings**
>
> Thank you for this helpful comment. We have reorganized the paper so that the three components now form a clear, sequential argument. While we are preparing the full revision of the paper, we present the revised storyline in the official comment above and in the uploaded **Supplementary Material**, as well as in this [overview figure](https://ibb.co/3ymF9M0J).
>
> In brief, we begin by grounding our work in the framework of the NeuroAI Turing Test, which holds that a model is brain-aligned only if its alignment score is indistinguishable from the distribution of inter-subject alignment scores. Our first contribution provides the first large-scale, ROI-resolved application of this idea to naturalistic video fMRI.
>
> We then extend the idea of distributional assessment to **model rankings**, finding many of them to be practically equivalent in terms of brain-alignment despite differing in architecture, training objective and/or dataset.
>
> These limitations of standard benchmarking pipelines in terms of discriminatory power lead directly to our **main conceptual contribution: alignment pattern similarity (APS) as a relational criterion to further gauge brain-alignment of models**. In addition to the *distributional criterion* discussed above, we introduce APS to assess whether a model reproduces the cross-region alignment pattern for a given brain region. Our premise is that a model should only be considered aligned to a brain region if it preserves how that region relates to other regions.
> We first show that this pattern is highly consistent across subjects. Applying APS to the top-performing models in our benchmark, we find that all models fall far short of reproducing the cross-region alignment patterns: brain-brain alignment patterns are highly similar to each other, but different from model-brain alignment patterns. Moreover, APS **allows to distinguish further** between models that result as practically equivalent according to RSA score alone.
>
> New results figures: [distribution of alignment pattern similarities](https://ibb.co/gZgHrqP9); [alignment patterns](https://ibb.co/W4jgwxm5)
>
> **Practical equivalence - clarification**
>
> We thank the reviewer for this question and the opportunity to clarify the methodology behind our practical equivalence analysis.
> We think there is a misunderstanding that we would like to clarify. We will adopt the terminology/notation suggested by Feather et al (NeuroAI Turing test).
> Our definition of practical equivalence of two models $m, n$ rests on comparing the distributions of their average alignment scores $\langle \mathcal{M} (\phi_m, \psi_i)\rangle_i $ and $\langle \mathcal{M} (\phi_n, \psi_i)\rangle_i $. Importantly, these are not subject-subject comparisons. We realise that our phrasing in terms of “within subject-level variability” (abstract) and “subject-level variability” (Methods) was misleading. We will rephrase this in the revised version of the paper. We provide a PDF with the updated Methods section in the uploaded **Supplementary Material**.
>
> Concerning the reviewer's question about the sensitivity of this analysis to the brain-brain baseline, we hope that this question is resolved by way of our clarification that the analysis actually does not rely on brain-brain alignment score distributions. We apologise for the confusion.
>
> **Conceptual differences and overlapping assumptions with NeuroAI Turing test**
>
> We appreciate this question, as it helped to refine and streamline our story. We now explicitly position our contributions relative to the NeuroAI Turing test, as outlined above.

---

### Author Response · Authors · 2025-11-26
**Reconceptualisation of alignment pattern analysis and new results**

We would like to thank all reviewers for their appreciation of our paper as well as their valuable feedback! We reply to each reviewer’s individual points below.

We would like to clarify the revised throughline of our paper, which includes a significant **conceptual improvement of the alignment pattern analysis** as well as an **explicit positioning relative to the NeuroAI Turing test**[1], illustrated in **[this overview figure](https://ibb.co/3ymF9M0J)**. Crucially, the alignment pattern analysis now directly compares brain-brain alignment patterns (derived from fMRI-data) and model-brain alignment patterns (instead of comparing each to the connectivity-derived alignment pattern). This reconceptualisation connects the idea presented in the NeuroAI Turing test [1] - that models should “generate internal representations that are indistinguishable from those recorded from humans” - to our relational criterion: **models should be indistinguishable from other brains in their alignment patterns**.
We additionally argue that the field should distinguish between **brain-aligned** models (which should satisfy the relational criterion) and **brain-predictive** models (which do not need to satisfy the relational criterion).

A more elaborate version of this reconceptualisation can be found in the uploaded **Supplementary Material**.

We have also **strengthened the quantification of our results** in three important ways.

1. We now estimate the **variance of brain-brain alignment patterns** and alignment pattern similarities (APS) by generating a distribution of alignment patterns between pairs of subjects. This defines the attainable brain-brain APS range to which model-brain APS can be compared, thereby **combining distributional [1] and relational criteria** for brain-alignment.  **Figures**: [distribution of alignment pattern similarities](https://ibb.co/gZgHrqP9);  [alignment patterns](https://ibb.co/W4jgwxm5);

2. We add a **model-brain APS baseline by estimating APS for randomly initialized networks**. This allows to separate the effects of model architecture from the effects of training objective and data on brain alignment. **Figures**: [distribution of APS trained vs. untrained](https://ibb.co/jPX3pbCF); [alignment patterns for randomly initialised models](https://ibb.co/bjVszGz1).

3. We are in the process of rerunning the original analysis, replacing our minimal connectivity graph with **structural connectivity data from the [Human Connectome Project](https://brainlife.io/pub/640a3f9dc538c16a826f9b1a?utm_source=chatgpt.com#release.1.preprocessing)**.

Note: All results presented are based on using RSA as alignment measure. We are currently generating results for Linear Predictivity. Models shown for each ROI are always the *n practically equivalent* models for that ROI resulting from the standard benchmarking pipeline.

[1]: Feather, Jenelle, et al. "Brain-model evaluations need the neuroai turing test." arXiv preprint arXiv:2502.16238 (2025).

---

### Author Response · Authors · 2025-12-03
**Summary of our paper, reviews, and revisions**

We would like to provide a summary of our paper in its revised form, as well as of the original reviews and how we addressed the reviewers' concerns.


**Summary of our paper**
We examine what it means for a vision model to be “brain-aligned”. Building on the NeuroAI Turing Test [1], which treats a model as aligned if its alignment scores are indistinguishable from brain–brain scores, we extend and refine existing benchmarks.

**Our contributions are threefold**:

1. **Comprehensive benchmarking**: We evaluate state-of-the-art models on a naturalistic video fMRI dataset spanning the visual hierarchy. Many models pass the Turing Test under Linear Predictivity (LP) but fail under Representational Similarity Analysis (RSA), showing alignment conclusions depend strongly on the metric.
2. **Practical equivalence across models**: Using a distributional criterion, we find that models differing in architecture, training data, and objectives often appear **practically equivalent in alignment** for most visual ROIs, revealing limited discriminative power of current benchmarks.
3. **Relational criterion for alignment**: We propose a relational criterion based on cross-region **alignment patterns (APs)** as a necessary criterion for brain alignment. Estimated from fMRI data, APs are highly consistent across subjects and characteristic of different brain regions. Many equivalently aligned models fail to reproduce these patterns.

We propose a distinction between strictly brain-aligned models and merely brain-predictive models that achieve high scores without capturing brain-like representational structure.

**Original reviews**

All reviewers highlighted the *relevance and timeliness* of evaluating brain-alignment benchmarks, the *soundness and rigour* of our analyses, and the *comprehensiveness of our benchmark*.

Their concerns were mainly about *clarity and presentation* and a *lack of controls/baselines for alignment pattern similarity (APS)*.
We addressed these as follows:
- **Figure quality**: Improved labels, resolution, and figure-text cohesion.
- **Conceptual throughline, connection between contributions, and reference to previous work**: Revised to explicitly connect our three contributions and clarify relation to the NeuroAI Turing Test [1] now supported by Fig. 1.
- **Interpretation of APS**: Clarified APS as necessary but not sufficient for brain alignment. Show RSA-based fMRI-derived APs track connectivity-derived APs, now estimated from a large DTI dataset.
- **Overall clarity**: We have improved the clarity of the paper, including the introduction of consistent notation for APs.
- **Controls, baselines and variance estimates for APS**: For each type of AP, we introduced a control:
  - fMRI-derived APs: We estimate a distribution of fMRI-derived APs based on pairwise predictions between subjects. This allows us to quantify the consistency of APs across subjects.
  - model-derived APs: As suggested by the reviewers, we estimate AP and APS for untrained models as a baseline, finding that training strongly boosts RSA-based APS, but not LP-based APS.
  - connectivity-derived APs: We generate a null distribution of APS between connectivity-derived  and fMRI-derived APs by sampling 100 random connectivity-derived APs.

*Future work*

The reviewers suggested some more interesting analyses that we leave for future work .
- **more alignment measures and more fMRI datasets**: Reviewer NsfX suggested to repeat alignment pattern similarity for more metrics and other fMRI datasets. While these are important and interesting points, we decided to leave this for future work as our main contribution is the introduction of APS as a concept, and we think evaluating on more metrics might detract from the clarity of the storyline, and evaluating on more fMRI datasets is out of the scope of the current work.
- **layer-wise APS**: Reviewer NsfX suggested to do layer-wise APS, which we understand amounts to using APS as a primary alignment measure. As we now clarify, we intend APS to be used as a complementary criterion, i.e. after selecting the best layer according to the primary alignment measure.
- **alignment pattern similarity between layers of a model**: Reviewer yHdD suggested to apply APS to layers of a model. This is an interesting idea, in a similar spirit as previous work by Nonaka et al. [2]. However, as this would have required extensive new experiments and as we are worried that including this might detract from the clarity of the paper, we leave this for future work.

**With these revisions, we are confident that our manuscript now offers a robust, rigorous, and clearly articulated framework that will guide the next generation of brain-aligned vision modeling.**

[1] Feather et al. (2025)

[2] Nonaka et al. (2021)

---

### Meta-Review · Area_Chair_11iu · 2026-01-02

**Summary:**

The reviewers broadly agreed that the paper addresses an important and timely question: how to meaningfully evaluate model-brain alignment beyond single-metric rankings such as RSA or linear predictivity. Reviewers found the large-scale benchmarking on a naturalistic video fMRI dataset to be solid and the practical-equivalence analysis useful in highlighting the limited discriminative power of current evaluation pipelines. The main point of discussion concerned the proposed alignment pattern similarity (APS) criterion: while reviewers found the idea novel and potentially impactful, several raised concerns about clarity, validation, and interpretation, as well as about how clearly APS was motivated by and connected to the earlier benchmarking results.

**Reviewer Concerns:**

The rebuttal appropriately addressed several concerns raised by the reviewers:
- The authors clarified the overall narrative by explicitly positioning the paper as an extension of distributional ideas from the NeuroAI Turing Test, and by motivating APS as a relational criterion that follows from the limited discriminative power of ranking-based benchmarks and practical equivalence analyses.
- The interpretation of APS was clarified as a necessary but not sufficient condition for brain alignment, addressing concerns that the criterion might be overly strict or philosophically inappropriate.
- Additional analyses and controls were added or described, including distributions of brain-brain alignment patterns, random-weight model baselines for APS, and comparisons to structural connectivity, strengthening the empirical grounding of the APS results.
- Several sources of confusion around baselines and subject variability were clarified, particularly regarding how practical equivalence is defined and evaluated.

However, some concerns remain:
- The evaluation continues to focus primarily on RSA and linear predictivity, despite the paper's broader critique of alignment metrics; while this is acknowledged and partially justified, some reviewers may still view the scope as limited.
- Some suggested extensions (e.g., additional metrics or datasets) are deferred to future work.

Overall, while not all concerns are fully resolved, the main technical and conceptual misunderstandings raised in the reviews were largely addressed.

**Reviewer Scores:**

- jG2B raised concerns primarily about conceptual coherence, baselines, and positioning relative to prior work (including the NeuroAI Turing Test). These issues were directly addressed in the rebuttal through restructuring and clarification, and it is likely that this reviewer's score would flip to a 6 with the revised framing and added analyses.
- NsfX expressed broader skepticism about metric choice, APS validation, and interpretability. While the rebuttal added relevant baselines and clarified the intended scope of APS, this reviewer may still view the paper as limited in scope and stick to their 4.
- yHdD was generally positive about the novelty and importance of the contribution, while raising philosophical and clarity-related concerns. The rebuttal engaged directly with these points, and this reviewer's score would likely remain at a 6.

---

### Decision · Program_Chairs · 2026-01-26

Accept (Poster)